# WoodFisher: Efficient Second-Order Approximation for Neural Network Compression

**Sidak Pal Singh**[*]
ETH Zurich, Switzerland
contact@sidakpal.com

**Dan Alistarh**
IST Austria & Neural Magic, Inc.
dan.alistarh@ist.ac.at

## Abstract

Second-order information, in the form of Hessian- or Inverse-Hessian-vector products, is a fundamental tool for solving optimization problems. Recently, there has been significant interest in utilizing this information in the context of deep neural networks; however, relatively little is known about the quality of existing approximations in this context. Our work examines this question, identifies issues with existing approaches, and proposes a method called WoodFisher to compute a faithful and efficient estimate of the inverse Hessian.

Our main application is to neural network compression, where we build on the classic Optimal Brain Damage/Surgeon framework. We demonstrate that WoodFisher significantly outperforms popular state-of-the-art methods for one-shot pruning. Further, even when iterative, gradual pruning is allowed, our method results in a gain in test accuracy over the state-of-the-art approaches, for standard image classification datasets such as ImageNet ILSVRC. We examine how our method can be extended to take into account first-order information, as well as illustrate its ability to automatically set layer-wise pruning thresholds and perform compression in the limited-data regime. The code is available at the following link, https://github.com/IST-DASLab/WoodFisher.

## 1   Introduction

The recent success of deep learning, e.g. [1, 2] has brought about significant accuracy improvement in areas such as computer vision [3, 4] or natural language processing [5, 6]. Central to this performance progression has been the size of the underlying models, with millions or even billions of trainable parameters [4, 5], a trend which seems likely to continue for the foreseeable future [7].

Deploying such large models is taxing from the performance perspective. This has fuelled a line of work where researchers compress such parameter-heavy deep neural networks into "lighter," easier to deploy variants. This challenge is not new, and in fact, results in this direction can be found in the early work on neural networks, e.g. [8–10]. Thus, most of the recent work to tackle this challenge can find its roots in these classic references [11], and in particular in the Optimal Brain Damage/Surgeon (OBD/OBS) framework [8, 10]. Roughly, the main idea behind this framework is to build a local quadratic model approximation based on the second-order Taylor series expansion to determine the optimal set of parameters to be removed. (We describe it precisely in Section 4.)

A key requirement to apply this approach is to have an accurate estimate of the inverse Hessian matrix, or at least to accurate inverse-Hessian-vector-products (IHVPs). In fact, IHVPs are a central ingredient in many parts of machine learning, most prominently for optimization [12–15], but also in other applications such as influence functions [16] or continual learning [17]. Applying second-order methods at the scale of model sizes described above might appear daunting, and so is often done via

---

[*]Work done while at IST Austria.

coarse-grained approximations (such as diagonal, block-wise, or Kronecker-factorization). However, relatively little is understood about the quality and scalability of such approximations.

**Motivation.** Our work centers around two main questions. The first is analytical, and asks if second-order approximations can be both *accurate* and *scalable* in the context of neural network models. The second is practical, and concerns applications of second-order approximations to neural network compression. In particular, we investigate whether these methods can be competitive with both industrial-scale methods such as *magnitude-based pruning* [18], as well as with the series of non-trivial compression methods proposed by researchers over the past couple of years [19–24].

**Contribution.** We first examine second-order approximation schemes in the context of convolutional neural networks (CNNs). In particular, we identify a method of approximating Hessian-Inverse information leveraging the structure of the empirical Fisher information matrix to approximate the Hessian, in conjunction with the Woodbury matrix identity to provide iteratively improving approximations of Inverse-Hessian-vector products. We show that this method, which we simply call WoodFisher, can be computationally-efficient, and that it faithfully represents the structure of the Hessian even for relatively low sample sizes. We note that early variants of this method have been considered previously [10, 25], but we believe we are the first to consider its accuracy, efficiency, and implementability in the context of large-scale deep models, as well as to investigate its extensions.

To address the second, practical, question, we demonstrate in Section 4 how WoodFisher can be used in conjunction with variants of the OBD/OBS pruning framework, resulting in state-of-the-art compression of popular convolutional models such as ResNet50 and MobileNet on the ILSVRC (ImageNet) dataset [26]. We investigate two practical application scenarios.

The first is *one-shot* pruning, in which the model has to be compressed in a single step, without any re-training. Here, WoodFisher easily outperforms all previous methods based on approximate second-order information or global magnitude pruning. The second scenario is *gradual* pruning, allowing for re-training between pruning steps. Surprisingly, even here WoodFisher either matches or outperforms state-of-the-art pruning approaches, including recent dynamic pruners [24, 27]. Our study focuses on *unstructured* pruning, but we can exhibit non-trivial speedups for real-time inference by running on a CPU framework which efficiently supports unstructured sparsity [28].

WoodFisher has several useful features and extensions. Since it approximates the full Hessian inverse, it can provide a *global* measure of parameter importance, and therefore removes the need for manually choosing sparsity targets per layer. Second, it allows us to apply compression in the limited-data regime, where either e.g. 99% of the training is unavailable, or no data labels are available. Third, we show that we can also take into account the first-order (gradient) term in the local quadratic model, which leads to further accuracy gain, and the ability to prune models which are not fully converged.

## 2 Background

**Deterministic Setting.** We consider supervised learning, where we are given a training set $S = \{(\mathbf{x}_i, \mathbf{y}_i)\}_{i=1}^N$, comprising of pairs of input examples $\mathbf{x} \in \mathcal{X}$ and outputs $\mathbf{y} \in \mathcal{Y}$. The goal is to learn a function $f : \mathcal{X} \mapsto \mathcal{Y}$, parametrized by weights $\mathbf{w} \in \mathbb{R}^d$, such that given input $\mathbf{x}$, the prediction $f(\mathbf{x}; \mathbf{w}) \approx \mathbf{y}$. We consider the loss function $\ell : \mathcal{Y} \times \mathcal{Y} \mapsto \mathbb{R}$ to measure the accuracy of the prediction. The training loss $L$ is defined as the average over training examples, i.e., $L(\mathbf{w}) = \frac{1}{N} \sum_{n=1}^N \ell(\mathbf{y}_n, f(\mathbf{x}_n; \mathbf{w}))$.

**The Hessian Matrix.** For a twice differentiable loss $L$, the Hessian matrix $\mathbf{H} = \nabla_{\mathbf{w}}^2 L$, takes into account the local geometry of the loss at a given point $\mathbf{w}$ and allows building a faithful approximation to it in a small neighbourhood $\delta \mathbf{w}$ surrounding $\mathbf{w}$. This is often referred to as the local quadratic model for the loss and is given by $L(\mathbf{w} + \delta \mathbf{w}) \approx L(\mathbf{w}) + \nabla_{\mathbf{w}} L^\top \delta \mathbf{w} + \frac{1}{2} \delta \mathbf{w}^\top \mathbf{H} \, \delta \mathbf{w}$.

**Probabilistic Setting.** An alternative formulation is in terms of the underlying joint distribution $Q_{\mathbf{x},\mathbf{y}} = Q_{\mathbf{x}} \, Q_{\mathbf{y}|\mathbf{x}}$. The marginal distribution $Q_{\mathbf{x}}$ is generally assumed to be well-estimated by the empirical distribution $\widehat{Q}_{\mathbf{x}}$ over the inputs in the training set. As our task is predicting the output $\mathbf{y}$ given input $\mathbf{x}$, training the model is cast as learning the conditional distribution $P_{\mathbf{y}|\mathbf{x}}$, which is close to the true $Q_{\mathbf{y}|\mathbf{x}}$. If we formulate the training objective as minimizing the KL divergence between these conditional distributions, we obtain the equivalence between losses $\ell(\mathbf{y}_n, f(\mathbf{x}_n; \mathbf{w})) = -\log(p_{\mathbf{w}}(\mathbf{y}_n|\mathbf{x}_n))$, where $p_{\mathbf{w}}$ is the density function corresponding to the model distribution.

**The Fisher Matrix.** In the probabilistic view, the Fisher information matrix $F$ of the model's conditional distribution $P_{y|x}$ is defined as,

$$F = E_{P_{\mathbf{x},\mathbf{y}}} \left[ \nabla_{\mathbf{w}} \log p_{\mathbf{w}}(\mathbf{x}, \mathbf{y}) \nabla_{\mathbf{w}} \log p_{\mathbf{w}}(\mathbf{x}, \mathbf{y})^{\top} \right] . \tag{1}$$

In fact, it can be proved that the Fisher $F = E_{P_{\mathbf{x},\mathbf{y}}} \left[ -\nabla_{\mathbf{w}}^2 \log p_{\mathbf{w}}(\mathbf{x}, \mathbf{y}) \right]$. Then, by expressing $P_{\mathbf{y},\mathbf{x}} = Q_{\mathbf{x}} P_{\mathbf{y}|\mathbf{x}} \approx \widehat{Q}_{\mathbf{x}} P_{\mathbf{y}|\mathbf{x}}$ and under the assumption that the model's conditional distribution $P_{\mathbf{y}|\mathbf{x}}$ matches the conditional distribution of the data $\widehat{Q}_{\mathbf{y}|\mathbf{x}}$, the Fisher and Hessian matrices are equivalent.

**The Empirical Fisher.** In practical settings, it is common to consider an approximation to the Fisher matrix introduced in Eq. (1), where we replace the model distribution $P_{\mathbf{x},\mathbf{y}}$ with the empirical training distribution $\widehat{Q}_{\mathbf{x},\mathbf{y}}$. Thus we can simplify the expression of empirical Fisher as follows,

$$\hat{F} = E_{\widehat{Q}_{\mathbf{x}}} \left[ E_{\widehat{Q}_{\mathbf{y}|\mathbf{x}}} \left[ \nabla \log p_{\mathbf{w}}(\mathbf{y}|\mathbf{x}) \nabla \log p_{\mathbf{w}}(\mathbf{y}|\mathbf{x})^{\top} \right] \right] \stackrel{(a)}{=} \frac{1}{N} \sum_{n=1}^{N} \underbrace{\nabla\ell\left(\mathbf{y}_n, f\left(\mathbf{x}_n; \mathbf{w}\right)\right)}_{\nabla \ell_n} \nabla\ell\left(\mathbf{y}_n, f\left(\mathbf{x}_n; w\right)\right)^{\top}$$

where (a) uses the equivalence of the loss between the probabilistic and deterministic settings. In the following, we will use a shorthand $\ell_n$ to denote the loss for a particular training example $(\mathbf{x}_n, \mathbf{y}_n)$, and refer to the Fisher as *true Fisher*, when needed to make the distinction relative to empirical Fisher.

## 3 Efficient Estimates of Inverse-Hessian Vector Products

Second-order information in the form of Inverse-Hessian Vector Products (IHVP) has several uses in optimization and machine learning [29, 30]. Since computing and storing the Hessian and Fisher matrices directly is prohibitive, we will focus on efficient ways to approximate this information.

As we saw above, the Hessian and Fisher matrices are equivalent if the model and data distribution match. Hence, under this assumption, the Fisher can be seen as a reasonable approximation to the Hessian. Due to its structure, the Fisher is positive semidefinite (PSD), and hence can be made invertible by adding a small diagonal dampening term. This approximation is therefore fairly common [15, 20, 31], although there is relatively little work examining the quality of this approximation in the context of neural networks.

Further, one can ask whether the *empirical Fisher* is a good approximation of the true Fisher. The latter is known to converge to the Hessian as the training loss approaches zero via relation to Gauss-Newton. The Empirical Fisher does not enjoy this property, but is far more computationally-efficient than the Fisher, as it can be obtained after a limited number of back-propagation steps. Hence, this second approximation would trade off theoretical guarantees for practical efficiency. In the next section, we examine how these approximations square off in practice for neural networks.

### 3.1 The (Empirical) Fisher and the Hessian: A Visual Tour

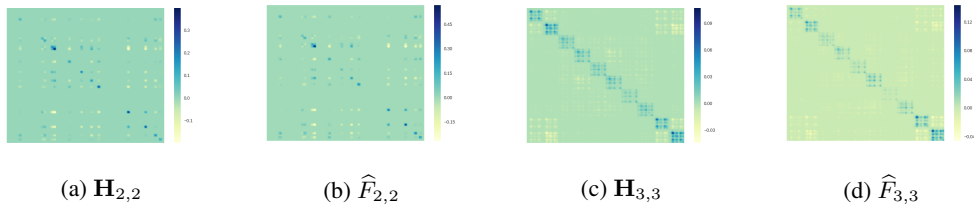

|     |     |     |     |
| --- | --- | --- | --- |
| (a) $\mathbf{H}_{2,2}$ | (b) $\widehat{F}_{2,2}$ | (c) $\mathbf{H}_{3,3}$ | (d) $\widehat{F}_{3,3}$ |

Figure 1: Hessian ($\mathbf{H}$) and empirical Fisher ($\widehat{F}$) blocks for CIFARNET ($3072 \rightarrow 16 \rightarrow 64 \rightarrow 10$) corresponding to second and third hidden layers when trained on CIFAR10. Figures have been smoothened slightly with a Gaussian kernel for better visibility. Both Hessian and empirical Fisher have been estimated over a batch of 100 examples in all the figures.

We consider the Hessian ($\mathbf{H}$) and empirical Fisher ($\hat{F}$) matrices for neural networks trained on standard datasets like CIFAR10 and MNIST. Due to practical considerations, we consider relatively small models: on CIFAR10, we consider a fully-connected network with two hidden layers $3072 \rightarrow 16 \rightarrow 64 \rightarrow 10$, which we refer to as CIFARNET.

Figure 1 compares the Hessian and empirical Fisher blocks corresponding to the second and third hidden layers of this network. Visually, there is a clear similarity between the structure of these two matrices, for both the layers. A similar trend holds for the first hidden layer as well as the cross-layer blocks, and for MNIST examples. Surprisingly, this behaviour occurs even if the network is not at full convergence, where we would expect the data and model distribution to match, but even at early stages of training (e.g., after one epoch of training). (Please see Appendix S3 for full experiments.) This observation is consistent with recent work [32] finding high cosine similarity between the Hessian and empirical Fisher matrices just after a few gradient steps.

As can be noted from the Figure 1, the main difference between these matrices is not in terms of structure, but in terms of *scale*. As a result, we could consider that the empirical Fisher $\hat{F} \propto \mathbf{H}$, modulo scaling, as long as our target application is not scale-dependent, or if we are willing to adjust the scaling through hyper-parametrization. Assuming we are willing to use the empirical Fisher as a proxy for the Hessian, the next question is: how can we estimate its inverse efficiently?

### 3.2 The WoodFisher Approximation

**The Woodbury Matrix Identity.** Clearly, direct inversion techniques would not be viable, since their runtime is cubic in the dimension parameter. Instead, we start from the Woodbury matrix identity[2], providing the formula for computing the inverse of a low-rank correction to a given invertible matrix $A$. The Sherman-Morrison formula is a simplified variant, given as $\left(A + \mathbf{u}\mathbf{v}^\top\right)^{-1} = A^{-1} - \frac{A^{-1}\mathbf{u}\mathbf{v}^\top A^{-1}}{1+\mathbf{v}^\top A^{-1}\mathbf{u}}$. We can express the empirical Fisher as the recurrence,

$$\widehat{F}_{n+1} = \widehat{F}_n + \frac{1}{N}\nabla\ell_{n+1}\nabla\ell_{n+1}^\top, \quad \text{where} \quad \widehat{F}_0 = \lambda I_d. \tag{2}$$

Above, $\lambda$ denotes the *dampening* term, i.e., a positive scalar $\lambda$ times the identity $I_d$ to render the empirical Fisher positive definite. Then, the recurrence for calculating the inverse of empirical Fisher becomes:

$$\widehat{F}_{n+1}^{-1} = \widehat{F}_n^{-1} - \frac{\widehat{F}_n^{-1}\nabla\ell_{n+1}\nabla\ell_{n+1}^\top\widehat{F}_n^{-1}}{N + \nabla\ell_{n+1}^\top\widehat{F}_n^{-1}\nabla\ell_{n+1}}, \quad \text{where} \quad \widehat{F}_0^{-1} = \lambda^{-1}I_d. \tag{3}$$

Finally, we can express the inverse of the empirical Fisher as $\widehat{F}^{-1} = \widehat{F}_{N+1}^{-1}$. Stretching the limits of naming convention, we refer to this method of using the empirical Fisher in place of Hessian and computing its inverse via the Woodbury identity as *WoodFisher*.

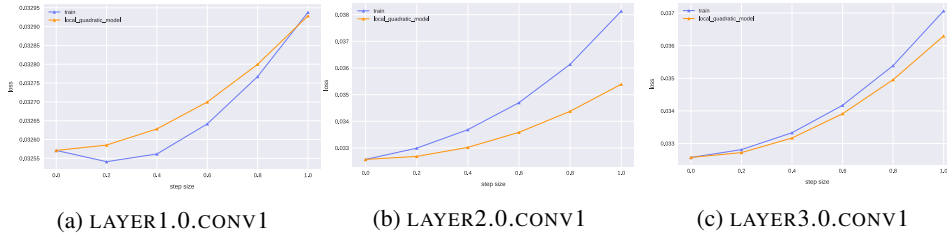

(a) LAYER1.0.CONV1      (b) LAYER2.0.CONV1      (c) LAYER3.0.CONV1

Figure 2: Approximation quality of the loss suggested by the local quadratic model using WoodFisher with respect to the actual training loss. The three plots measure the quality of local quadratic model along three different directions, each corresponding to pruning the respective layers to $50\%$ sparsity.

**Approximation quality for the local quadratic model.** To evaluate the accuracy of our local quadratic model estimated via WoodFisher, we examine how the loss predicted by it compares against the actual training loss. (Since we use a pre-trained network, the first-order gradient term is ignored; we will revisit this assumption later.) We test the approximation on three different directions, each corresponding to the pruning direction (of the form $\mathbf{H}^{-1}\delta\mathbf{w}$) obtained when compressing a particular layer to $50\%$ sparsity. We choose three layers from different stages of a pre-trained RESNET-20 on CIFAR10, and Figure 2 presents these results.

In all three cases, the local quadratic model using WoodFisher predicts an accurate approximation to the actual underlying loss. Further, it is possible to use the dampening $\lambda$ to control whether a more conservative or relaxed estimate is needed. Overall, this suggests that the approach might be fairly accurate, a hypothesis we examine in more detail in Section 4.

**Computational Efficiency and Block-wise Approximation.** While the expression in Eq. (3) goes until $n = N$, in practice, we found the method only needs a small subset of examples, $m$, typically ranging between $100$ to $400$. The runtime is this reduced from cubic to quadratic in $d$, which can still be excessive for neural networks with millions of parameters.

Thus, for large models we will need to employ a *block-wise approximation*, whereby we maintain and estimate limited-size blocks ('chunks') on the diagonal and ignore the off-diagonal parts. This block-wise simplification is is motivated by the observation that Hessians tend to be diagonally-dominant, and has been employed in previous work, e.g. [33]. Assuming uniform blocks of size $c \times c$ along the diagonal, the runtime of this inversion operation becomes $\mathcal{O}(mcd)$, and hence linear in the dimension $d$. This restriction appears necessary for computational tractability.

## 3.3 Context and Alternative Methods

There is a large body of work utilizing second-order information in machine learning and optimization, to the extent that, it is infeasible to discuss every alternative in detail here. We therefore highlight the main methods for estimating inverse Hessian-vector products (IHVPs) in our context of neural networks. See Appendix S2 for detailed discussion.

A tempting first approach is the *diagonal approximation*, which only calculates the elements along the diagonal, and inverts the resulting matrix. Variants of this approach have been employed in optimization [13, 14] and model compression [20]. Yet, as we show experimentally (Figure 3a), this local approximation can be surprisingly inaccurate. By comparison, WoodFisher costs an additional constant factor, but provides significantly better IHVP estimates. *Hessian-free* methods are another approach, which forgoes the explicit computation of Hessians [34] in favour of computing IHVP with a vector $\mathbf{v}$ by solving the linear system $\mathbf{Hx} = \mathbf{v}$ for some given $\mathbf{x}$. Unfortunately, a disadvantage of these methods, which we observed practically, is that they require many iterations to converge, since the underlying Hessian matrix can be ill-conditioned. *Neumann-series-based methods* [35, 36] exploit the infinite series expression for the inverse of a matrix with eigenspectrum in $[0, 1]$. This does not hold by default for the Hessian, and requires using the Power method to estimate the largest and smallest absolute eigenvalues, which increases cost substantially, while the Power method may fail to return the smallest negative eigenvalue.

**K-FAC.** Kronecker-factorization (K-FAC) methods [15, 37] replace the expectation of a Kronecker product between two matrices (that arises in the formulation of Fisher blocks between two layers) as the Kronecker product between the expectations of two matrices. This is known to be a significant approximation [15]. The main benefit of K-FAC is that the inverse can be efficiently computed [21, 22, 38, 39]. However, a significant drawback is that the Kronecker factorization form only exists naturally for fully-connected networks. When applied to convolutional or recurrent neural networks, the Kronecker structure needs to make further approximations [40, 41], limiting its applicability. Furthermore, even in regards to its efficiency, often approximations like the chunking the layer blocks or channel-grouping are required [42]. Also in Figure 3b, we show that when used for pruning WoodFisher can outperform K-FAC, even for fully-connected networks.

**WoodFisher.** In this context, with WoodFisher, we propose a new method of estimating second-order information that addresses some of the shortcomings of previous methods, and validate it in the context of network pruning. We emphasize that a similar approach was used in the early works of [10, 25] for the case of 1-hidden layer neural network with $< 100$ parameters. Our main contribution is in significantly extending this idea by scaling it to modern network sizes and examining the approximation relative to recent techniques (besides, other contributions like WoodTaylor, Section 6).

The Woodbury matrix identity was also used in L-OBS [33] by defining separate layer-wise objectives, and was applied to carefully-crafted blocks at the level of neurons. Our approach via empirical Fisher is more general, and we show experimentally that it yields better approximations at scale (Figure S1).

**Use of Empirical Fisher.**  Kunstner et al. [30] questioned the use of empirical Fisher since, as the training residuals approach zero, the empirical Fisher goes to zero while the true Fisher approaches the Hessian. However, this rests on the assumption that each individual gradient vanishes for well-optimized networks, which we did not find to hold in our experiments. Further, they argue that a large number of samples are needed for the empirical Fisher to serve as a good approximation—in our experiments, we find that a few hundred samples suffice for our applications (e.g. Figure 1).

## 4  Model Compression

This area has seen an explosion of interest in recent years–due to space constraints, we refer the reader to the recent survey of [11] for an overview, and mainly focus on closely related work on *unstructured* pruning. Broadly, existing methods can be split into four classes: (1) methods based on approximate second-order information, e.g. [20, 22, 33], usually set in the classical OBD/OBS framework [8, 10]; (2) iterative methods, e.g. [19, 43, 44], which apply magnitude-based weight pruning in a series of incremental steps over fully- or partially-trained models; (3) dynamic methods, e.g. [23, 24, 27, 45], which prune during regular training and can additionally allow the re-introduction of weights during training; (4) variational or regularization-based methods, e.g. [46, 47]. Recently, pruning has also been linked to intriguing properties of neural network training [48]. WoodFisher belongs to the first class of methods, but can be used together with both iterative and dynamic methods.

**Optimal Brain Damage.**  We start from the idea of pruning (setting to 0) the parameters which, when removed, lead to a minimal increase in training loss. Denote the dense weights by $\mathbf{w}$, and the new weights after pruning as $\mathbf{w} + \delta\mathbf{w}$. Using the local quadratic model, we seek to minimize $\delta L = L(\mathbf{w} + \delta\mathbf{w}) - L(\mathbf{w}) \approx \nabla_{\mathbf{w}} L^\top \delta\mathbf{w} + \frac{1}{2}\delta\mathbf{w}^\top \mathbf{H}\, \delta\mathbf{w}$. It is often assumed that the network is pruned at a local optimum, which eliminates the first term. (We revisit this in Section 6.)

If we consider the simple case where a single parameter, at index $q$, is removed, we get that corresponding optimal perturbation $\delta\mathbf{w}^*$ and change in loss $\delta L^*$ are, as detailed in Appendix S1.2:

$$\delta\mathbf{w}^* = \frac{-w_q \mathbf{H}^{-1}\mathbf{e}_q}{[\mathbf{H}^{-1}]_{qq}}, \quad \text{and} \quad \delta L^* = \frac{w_q^2}{2\,[\mathbf{H}^{-1}]_{qq}}. \tag{4}$$

Then, the best choice of $q$ corresponds to removing that parameter $w_q$ which has the minimum value for the change in loss $\delta L^*$, and we refer to this as the pruning statistic $\rho_q$. Extending this analysis to multiple parameters is combinatorially hard. Therefore, as an approximation, when removing multiple parameters, we sort the parameters by the pruning statistic $\rho_q$, removing those with the smallest values. The overall weight perturbation in such a scenario is computed by adding the optimal weight update, Eq. (4), for each parameter that we decide to prune. (We mask the weight update at the indices of removed parameters to zero, so as to adjust for adding the weight updates separately.) We call this resulting weight update the *the pruning direction*.

If the Hessian is assumed to be diagonal, we recover the pruning statistic of optimal brain damage [8], $\delta L_{\text{OBD}}^* = \frac{1}{2}w_q^2[\mathbf{H}]_{qq}$. Further, if we let the Hessian be isotropic, we obtain the case of *magnitude pruning*, one of the leading practical methods [44], as the statistic amounts to $\delta L_{\text{Mag}}^* = \frac{1}{2}w_q^2$.

**Pruning using WoodFisher.**  We use WoodFisher to get estimates of the Hessian inverse required in Eq. (4). Next, the decision to remove parameters based on their pruning statistic can be made either independently for every layer, or jointly across the whole network. The latter option allows us to automatically adjust the sparsity distribution across the various layers given a global sparsity target for the network. As a result, we do not have to perform a sensitivity analysis for the layers or use heuristics such as skipping the first or the last layers, as commonly done in prior work. We refer to the latter as *joint (or global)*-WoodFisher and the former as *independent (or layerwise)*-WoodFisher.

## 5  Experimental Results

We now apply WoodFisher for compressing CNNs on image classification tasks. We consider both *one-shot* and *gradual* pruning, and investigate how the Fisher sample size and block-wise assumptions affect the quality of the approximation, and whether this can lead to more accurate pruned models.

### 5.1 One-shot pruning.

Assume that we are given a pre-trained neural network which we would like to sparsify in a single step, without any re-training. This scenario might arise when having access to limited data, making re-training infeasible, and allows us to directly compare approximation quality.

**RESNET-20, CIFAR10.** First, we consider a pre-trained RESNET-20 [4] network on CIFAR10 with $\sim 300K$ parameters. We compute the inverse of the diagonal blocks corresponding to each layer. Figure 3a contains the test accuracy results for one-shot pruning in this setting, averaged across four seeds, as we increase the percentage of weights pruned. Despite the block-wise approximation, we observe that both independent- and joint-WoodFisher variants significantly outperform magnitude pruning and diagonal-Fisher based pruning.

We also compare against the global version of magnitude pruning, which can re-adjust sparsity across layers. Still, we find that the global magnitude pruning is worse than WoodFisher-independent until about 60% sparsity, beyond which it is likely that adjusting layer-wise sparsity is becomes essential. WoodFisher-joint performs the best amongst all the methods, and is better than the top baseline of global magnitude pruning - by about 5% and 10% in test accuracy at the 70% and 80% sparsity levels. Notice also the improvement relative to block size. Finally, diagonal-Fisher performs worse than magnitude pruning for sparsity levels higher than 30%. This finding was consistent, and so we omit it in the sections ahead. (We used 16,000 samples to estimate the diagonal Fisher, whereas WoodFisher performs well even with 1,000 samples.)

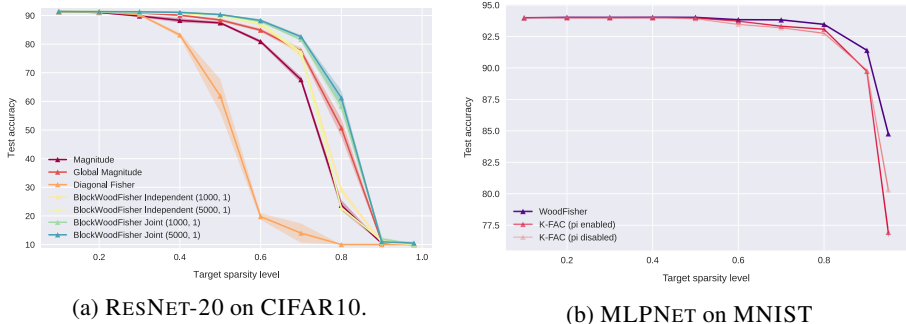

(a) RESNET-20 on CIFAR10.

(b) MLPNET on MNIST

Figure 3: One-shot pruning results of WoodFisher compared with: (a) magnitude and diagonal Fisher based pruning (b) K-FAC based pruning. Also, see the comparison against L-OBS [33] in Figure S1.

**Comparison against K-FAC.** We consider the scenario of one-shot pruning of MLPNET on MNIST. For both WoodFisher and K-FAC, we utilize a block-wise estimate of the Fisher (with respect to the layers, i.e., no further chunking). Figure 3b illustrates these results for the 'joint' pruning mode (however, similar results can be observed in the 'independent' mode too). The number of samples used for estimating the inverse is the same across K-FAC and WoodFisher (i.e., $50,000$ samples)[3]. This highlights the better approximation quality provided by WoodFisher, which unlike K-FAC does not make major assumptions. Note that, for convolutional layers, K-FAC needs to make additional approximations, so we can expect WoodFisher results to further improve over K-FAC.

**ResNet-50, IMAGENET.** We performed a similar experiment for the larger RESNET-50 model on ImageNet (25.5M parameters), which for efficiency we break into layer-wise blocks (chunks) of size 1K. We found that this suffices for significant performance gain over layer-wise and global magnitude pruning (as well as the L-OBS method [33]), as shown in Figure 4a. As a practical trick, we found it useful to replace individual gradients in the definition of the empirical Fisher with gradients averaged over a mini-batch of samples. Typically, we use 80 or 240 such averaged gradients over a mini-batch of size 100.

Additionally, Figure 4a , shows that the accuracy is further improved if we allow recomputation of the Hessian inverse estimate during pruning (but without retraining), as the local quadratic model

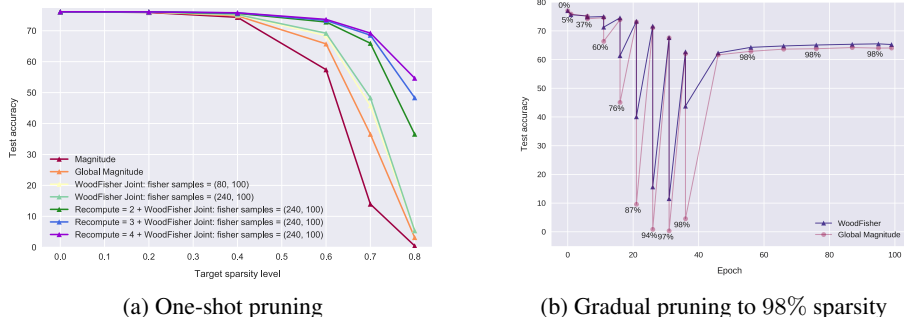

| (a) One-shot pruning | (b) Gradual pruning to 98% sparsity |

Figure 4: Results for **RESNET-50**, **IMAGENET**.

is valid otherwise only in a small neighbourhood or trust-region. The detailed results are present in Appendix S5.1, where we also show one-shot pruning results of MOBILENETV1 on IMAGENET, as well as ablation for the effect of chunk-size, dampening $\lambda$, # of samples used for Fisher computations.

## 5.2 Gradual Pruning

So far, the two best methods we identified in one-shot tests are WoodFisher (joint/global) and global magnitude pruning. We now compare these methods extensively against several previous methods. To facilitate comparison, we demonstrate our results on the pre-trained RESNET-50 and MOBILENETV1 models of the STR method [27], which claims state-of-the-art results. As in [27], all our IMAGENET experiments are run for 100 epochs on 4 NVIDIA V100 GPUs (i.e., $\sim$ 2.5 days for RESNET-50 and $\sim$ 1 day for MOBILENETV1). In terms of the pruning schedule, we follow the polynomial scheme of [19] (see illustration in Figure 4b right), and run WoodFisher and global magnitude in identical settings. See Appendix S4 for further details.

Table 1 presents our results with comparisons against numerous baselines for pruning RESNET-50 at the sparsity levels of 80%, 90%, and 95%. To take into account that some prior work uses different dense baselines, we also report the relative drop. WoodFisher outperforms all baselines, across both gradual and dynamic pruning approaches, in every sparsity regime. Compared to STR [27], Wood-Fisher improves accuracy at all sparsity lev-

|                   | Top-1 accuracy (%) | | Relative Drop | Sparsity |
|-------------------|----------|-----------|--------------------------------|----------|
| Method            | Dense ($D$) | Pruned ($P$) | $100 \times \frac{(P-D)}{D}$ | (%)      |
| DSR [49]          | 74.90    | 71.60     | -4.41                          | 80.00    |
| Incremental [19]  | 75.95    | 74.25     | -2.24                          | 73.50    |
| DPF [24]          | 75.95    | 75.13     | -1.08                          | 79.90    |
| GMP + LS [18]     | 76.69    | 75.58     | -1.44                          | 79.90    |
| VD [44, 46]       | 76.69    | 75.28     | -1.83                          | 80.00    |
| RIGL + ERK [45]   | 76.80    | 75.10     | -2.21                          | 80.00    |
| SNFS + LS [23]    | 77.00    | 74.90     | -2.73                          | 80.00    |
| STR [27]          | 77.01    | 76.19     | -1.06                          | 79.55    |
| Global Magnitude. | 77.01    | 76.60     | -0.53                          | 80.00    |
| DNW [50]          | 77.50    | 76.20     | -1.67                          | 80.00    |
| **WoodFisher**    | **77.01** | **76.73** | **-0.36**                     | **80.00** |
| GMP + LS [18]     | 76.69    | 73.91     | -3.62                          | 90.00    |
| VD [44, 46]       | 76.69    | 73.84     | -3.72                          | 90.27    |
| RIGL + ERK [45]   | 76.80    | 73.00     | -4.94                          | 90.00    |
| SNFS + LS [23]    | 77.00    | 72.90     | -5.32                          | 90.00    |
| STR [27]          | 77.01    | 74.31     | -3.51                          | 90.23    |
| Global Magnitude  | 77.01    | 75.09     | -2.49                          | 90.00    |
| DNW [50]          | 77.50    | 74.00     | -4.52                          | 90.00    |
| **WoodFisher**    | **77.01** | **75.26** | **-2.27**                     | **90.00** |
| GMP [18]          | 76.69    | 70.59     | -7.95                          | 95.00    |
| VD [44, 46]       | 76.69    | 69.41     | -9.49                          | 94.92    |
| VD [44, 46]       | 76.69    | 71.81     | -6.36                          | 94.94    |
| RIGL + ERK [45]   | 76.80    | 70.00     | -8.85                          | 95.00    |
| DNW [50]          | 77.01    | 68.30     | -11.31                         | 95.00    |
| STR [27]          | 77.01    | 70.40     | -8.58                          | 95.03    |
| Global Magnitude. | 77.01    | 71.65     | -6.96                          | 95.00    |
| **WoodFisher**    | **77.01** | **72.16** | **-6.30**                     | **95.00** |

Table 1: Comparing WoodFisher gradual pruning results with the state-of-the-art approaches. LS denotes label smoothing, ERK refers to Erdős-Renyi Kernel.

els, with a Top-1 test accuracy gain of $\sim 1\%$ and $1.7\%$ respectively at the 90% and 95% sparsity levels. The results averaged over multiple runs are similar and can be found in Appendix S5.4.

We also find that global magnitude (GM) is quite effective, surpassing many recent dynamic pruning methods, which also adjust the sparsity distribution across layers [24, 27, 45]. Comparing GM and WoodFisher, the latter outperforms at all sparsity levels, with higher gain at higher sparsities, e.g., $> 1\%$ boost in accuracy at 98% sparsity (see Table S4 of the Appendix). WoodFisher also outperforms Variational Dropout (VD) [46], the top-performing regularization-based method, on all

| Method | Top-1 accuracy (%) | | Relative Drop | Sparsity |
|---|---|---|---|---|
| | Dense ($D$) | Pruned ($P$) | $100 \times \frac{(P-D)}{D}$ | (%) |
| Incremental | 75.95 | 73.36 | -3.41 | 82.60 |
| SNFS | 75.95 | 72.65 | -4.34 | 82.00 |
| DPF | 75.95 | 74.55 | -1.84 | 82.60 |
| **WoodFisher** | 75.98 | **75.20** | **-1.03** | 82.70 |

Table 2: Comparison with state-of-the-art DPF [24] in a more commensurate setting by starting from a similarly trained dense baseline. The numbers for Incremental & SNFS are taken from [24].

sparsity targets, with a margin of $\sim 1.5\%$ at $80\%$ and $90\%$ sparsity. VD is also to be quite sensitive to initialization and hyperparameters [44], which can be partly seen from its results in the $95\%$ regime, where a $0.02\%$ difference in sparsity affects the obtained accuracy by over $2\%$.

Besides the comparison in Table 1, we further compare against another recent state-of-the-art DPF [24] in a more commensurate setting by starting from a similarly trained baseline. We follow their protocol and prune all layers except the last: see Table 2. In this setting as well, WoodFisher significantly outperforms DPF and the related baselines. The Appendix S5.3 contains additional experiments against the gradual magnitude pruning (GMP) baseline of [44], and on MOBILENETV1, against STR and Global Magnitude. We find that WoodFisher provides higher accuracy across all these cases.

To sum up, results show that WoodFisher outperforms state-of-the-art approaches, from each class of pruning methods, in all the considered sparsity regimes, setting a new state-of-the-art in unstructured pruning for these common benchmarks. The rationale behind its performance is provided in Figure 4b, showing how the methods behave during the course of gradual pruning. After almost every pruning step, WoodFisher provides a better pruning direction, and even with substantial retraining in between and after, global magnitude fails to catch up in terms of accuracy. This shows the benefit of using the second order information via WoodFisher to perform superior pruning steps.

**FLOPs and Inference Costs.** It is interesting to consider the actual speedup which can be obtained via these methods, as the total theoretical FLOP counts can be lower for methods such as STR. For this, we use the inference framework of [28], which supports the efficient execution of unstructured sparse convolutional models on CPUs. At batch size 1 (real-time inference), the dense baseline executes ResNet50 in 7.1 ms, whereas the STR $87\%$-pruned model executes in 4.1 ms, with Top-1 $74.3\%$ accuracy. By contrast, the WoodFisher uniformly-pruned model at $90\%$ executes in 4.3 ms, with accuracy $75.23\%$. For the same models at batch size 64, the times are: Dense $= 296$ ms, STR $= 146$ ms, and WoodFisher $= 157$ ms. Thus, at a relatively minor increase in inference time, there is a higher accuracy gain with WoodFisher models. Full results are given in the Appendix S6.

## 6    Discussion

**Extensions.** *(i) WoodTaylor: Pruning at a general point:* Incorporating the first-order gradient term in the Optimal Brain Damage framework should result in a more faithful estimate of the pruning direction, as many times in practice, the gradient is not exactly zero. Or it might be that pruning is being performed during training like in dynamic pruning methods. Hence, we redo the analysis by accounting for the gradient (see Appendix S9) and we refer to this resulting method as *'WoodTaylor'*. Note, an advantage of dynamic pruning methods is that the pruning is performed during training itself, although we have seen in Table 1, better results are obtained when pruning is performed post-training. Current dynamic pruning methods like [24] prune via global magnitude, and a possible future work would be to use WoodTaylor instead. Figure S13 presents some early results in this context, where pruning a partially trained network, yields an accuracy gain of $\sim 5\%$ over global magnitude pruning.

*(ii) Unlabeled Data.* While empirical Fisher inherently uses the label information when computing gradients, it is possible to avoid that and instead use a single sample from the model distribution, thus making it applicable to unlabeled data. (Appendix S7 shows this does not impact the results much).

**Future Work.** A few of the many interesting directions to apply WoodFisher, include, e.g., structured pruning which is easily facilitated by the OBD framework [21], pruning popular models used in NLP like Transformers, providing efficient IHVP estimates for influence functions, etc.

**Conclusion.** In sum, our work revisits the theoretical underpinnings of neural network pruning, and shows that foundational work can be successfully extended to large-scale settings, yielding state-of-the-art results. We hope that our findings can provide further momentum to the investigation of second-order properties of neural networks, and be extended to applications beyond compression.

## Broader Impact

Our work provides a general method for estimating second-order information at the scale of neural networks, and applies it to obtain state-of-the-art results on model compression. Our aim in doing so is to improve the performance of such machine learning applications. We apply our method to image classification, but our methods could be extended to any applications of neural networks. Our work could enable new, highly-accurate compressed models reducing inference times and resources required. The impact of any such application, such as for instance in surveillance, would need to be analyzed on a case-by-case basis and goes back to broader questions about the applicability of machine learning.

### Acknowledgements

This project has received funding from the European Research Council (ERC) under the European Union's Horizon 2020 research and innovation programme (grant agreement No 805223 ScaleML). Also, we would like to thank Alexander Shevchenko, Alexandra Peste, and other members of the group for fruitful discussions.

## Footnotes

[2]We chose Woodbury over Sherman-Morrison for the naming, since its general form can be used for approximating the inverse of true-Fisher or Gauss-Newton, when they can be expressed as sum of outer-products of the Jacobian.

[3]For the sake of efficiency, when using $50,000$ samples in the case of WoodFisher, we utilize 1000 averaged gradients over a mini-batch size of $50$. But even otherwise, we notice similar gains over K-FAC.

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
