[Supplementary Material]

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

# Appendix

## Contents

# S1 Model Compression in the Optimal Brain Damage framework

## S1.1 Problem Formulation

We assume that we are given a neural network which has been trained to convergence. This network typically has a huge number of parameters, in the order of millions. In this over-parameterized regime, we can expect that there exists a neural network within the given large neural network, which has fewer parameters but still achieves similar performance in comparison to the latter [48].

While ideally, we would want to simply train a neural network of the right size from the outset, this is harder than it seems. So currently, we just focus on pruning (or removing) the parameters from the given large neural network. In fact, this route of obtaining compact neural networks might still be needed to compress pre-trained networks that are already available and need to be deployed, say at an edge device.

Further, we will look at the setting of unstructured pruning, where individual weights and biases of the network are removed at a time, rather than deleting all parameters of a neuron at once (as in structured pruning). In a way, unstructured pruning is more general than structured pruning since all the parameters of a neuron might eventually get removed.

Hence, in other words, the goal here with pruning is to modify the current parameters of our neural network in a way that sets more of them to zero, while maintaining the performance. We use the training loss to measure the performance, and would like the change in training loss to be minimal after such a modification of the parameters.

Let us denote all the parameters of neural network by the vector $\mathbf{w}$, so as to analyse the effect of pruning at a more generic level, without distinguishing between weights and biases. Next, let $\delta\mathbf{w}$ denote this perturbation we apply to our parameters. The training loss is denoted by $L : \mathbb{R}^d \mapsto \mathbb{R}$ where $d$ is the dimensionality or size of parameters $\mathbf{w}$, and is typically given by:

$$L(\mathbf{w}) = \frac{1}{N} \sum_{i=1}^{N} \ell\big(y_i, \ f(\mathbf{x}_i; \mathbf{w})\big). \tag{5}$$

Here, $(\mathbf{x}_i, y_i)$ are samples from the training set $S$ of size $N$ and $f$ denotes the output of the neural network with parameters $\mathbf{w}$ at input $\mathbf{x}_i$.

Assuming that our update $\delta\mathbf{w}$ to the parameters is small enough, we can use the Taylor series expansion to measure the training loss at the new parameters $\mathbf{w} + \delta\mathbf{w}$. Consider, the second order Taylor series expansion of the function $L$ near $\mathbf{w}$ as follows:

$$L(\mathbf{w} + \delta\mathbf{w}) = L(\mathbf{w}) + \nabla_{\mathbf{w}}L^\top \delta\mathbf{w} + \frac{1}{2}\delta\mathbf{w}^\top \nabla_{\mathbf{w}}^2 L \ \delta\mathbf{w} + O\left(\|\delta\mathbf{w}\|^3\right) \tag{6}$$

To simplify notation, let us denote the hessian $\nabla_{\mathbf{w}}^2 L$ by $\mathbf{H}$ and the change in loss $L(\mathbf{w} + \delta\mathbf{w}) - L(\mathbf{w})$ by $\delta L$. Therefore, this change in loss can then be approximated as follows:

$$\delta L \approx \nabla_{\mathbf{w}}L^\top \delta\mathbf{w} + \frac{1}{2}\delta\mathbf{w}^\top \mathbf{H} \ \delta\mathbf{w} \tag{7}$$

## S1.2 Pruning at local optimum

In this section, we assume that network is pruned at (or close to) a local optimum. Hence, we can consider $\nabla_{\mathbf{w}}L = 0$ (or when close to a local optimum, $\nabla_{\mathbf{w}}L \approx 0$) and simplify the expression in Eq. (8) to the following:

$$\delta L \approx \frac{1}{2}\delta\mathbf{w}^\top \mathbf{H} \ \delta\mathbf{w} \tag{8}$$

### S1.2.1 Removing a single parameter $w_q$

Before proceeding further, we would like to remark that the analysis which factors in the first-order gradient term is considered in the Section S9 ahead. Now, our goal in pruning is to remove parameters that do not change the loss by a significant amount.

For now, let us consider the case when just a single parameter at index $q$ is removed. The corresponding perturbation $\delta \mathbf{w}$ can be expressed by the constraint $\mathbf{e}_q^\top \delta \mathbf{w} + w_q = 0$, where $\mathbf{e}_q$ denotes the $q^{\text{th}}$ canonical basis vector. Then pruning can be formulated as finding the optimal perturbation that satisfies this constraint, and the overall problem can written as follows:

$$\min_{\delta \mathbf{w} \in \mathbb{R}^d} \left( \frac{1}{2} \delta \mathbf{w}^\top \mathbf{H} \, \delta \mathbf{w} \right), \quad \text{s.t.} \quad \mathbf{e}_q^\top \delta \mathbf{w} + w_q = 0. \tag{9}$$

In order to impose the best choice for the parameter to be removed, we can further consider the following constrained minimization problem.

$$\min_{q \in [d]} \left\{ \min_{\delta \mathbf{w} \in \mathbb{R}^d} \left( \frac{1}{2} \delta \mathbf{w}^\top \mathbf{H} \, \delta \mathbf{w} \right), \quad \text{s.t.} \quad \mathbf{e}_q^\top \delta \mathbf{w} + w_q = 0 \right\}. \tag{10}$$

However, let us first focus on the inner problem, i.e., the one from Eq. (9). As this is a constrained optimization problem, we can consider the Lagrange multiplier $\lambda$ for the constraint and write the Lagrangian $\mathcal{L}(\delta \mathbf{w}, \lambda)$ as follows,

$$\mathcal{L}(\delta \mathbf{w}, \lambda) = \frac{1}{2} \delta \mathbf{w}^\top \mathbf{H} \, \delta \mathbf{w} + \lambda \left( \mathbf{e}_q^\top \delta \mathbf{w} + w_q \right). \tag{11}$$

The Lagrange dual function $g(\lambda)$, which is the infimum of the Lagrangian in Eq. (11) with respect to $\mathbf{w}$, can be then obtained by first differentiating Eq. 11 and setting it to 0, and then substituting the obtained value of $\delta \mathbf{w}$. These steps are indicated respectively in Eq. (12) and Eq. (13) below.

$$\mathbf{H} \delta \mathbf{w} + \lambda \mathbf{e}_q = 0 \implies \delta \mathbf{w} = -\lambda \mathbf{H}^{-1} \mathbf{e}_q. \tag{12}$$

$$g(\lambda) = \frac{\lambda^2}{2} \mathbf{e}_q^\top \mathbf{H}^{-1} \mathbf{e}_q - \lambda^2 \mathbf{e}_q^\top \mathbf{H}^{-1} \mathbf{e}_q + \lambda w_q = -\frac{\lambda^2}{2} \mathbf{e}_q^\top \mathbf{H}^{-1} \mathbf{e}_q + \lambda w_q. \tag{13}$$

Now, maximizing with respect to $\lambda$, we obtain that the optimal value $\lambda^*$ of this lagrange multiplier as

$$\lambda^* = \frac{w_q}{\mathbf{e}_q^\top \mathbf{H}^{-1} \mathbf{e}_q} = \frac{w_q}{[\mathbf{H}^{-1}]_{qq}}. \tag{14}$$

The corresponding optimal perturbation, $\delta \mathbf{w}^*$, so obtained is as follows:

$$\delta \mathbf{w}^* = \frac{-w_q \mathbf{H}^{-1} \mathbf{e}_q}{[\mathbf{H}^{-1}]_{qq}}. \tag{15}$$

Finally, the resulting change in loss corresponding to the optimal perturbation that removes parameter $w_q$ is,

$$\delta L^* = \frac{w_q^2}{2 [\mathbf{H}^{-1}]_{qq}}. \tag{16}$$

Going back to the problem in Eq. (10), the best choice of $q$ corresponds to removing that parameter $w_q$ which has the minimum value of the above change in loss. We refer to this change in loss as the pruning statistic $\rho$, see Eq. (17), which we compute for all the parameters and then sort them in the descending order of its value.

$$\boxed{\rho_q = \frac{w_q^2}{2 [\mathbf{H}^{-1}]_{qq}}.} \tag{17}$$

### S1.2.2 Removing multiple parameters at once

For brevity, consider that we are removing two parameters, $q_1$ and $q_2$, without loss of generality. The constrained optimization corresponding to pruning can be then described as follows,

$$\min_{q_1 \in [d], \, q_2 \in [d]} \left\{ \min_{\delta \mathbf{w} \in \mathbb{R}^d} \left( \frac{1}{2} \delta \mathbf{w}^\top \mathbf{H} \, \delta \mathbf{w} \right), \quad \text{s.t.} \quad \mathbf{e}_{q_1}^\top \delta \mathbf{w} + w_{q_1} = 0, \; \mathbf{e}_{q_2}^\top \delta \mathbf{w} + w_{q_2} = 0, \right\}. \tag{18}$$

We can see how the search space for the best parameter choices $(q_1, q_2)$ explodes exponentially. In general, solving this problem optimally seems to be out of hand. Although, it could be possible that the analysis might lead to a tractable computation for the optimal solution. Unfortunately, as described ahead, this is not the case and we will have to resort to some approximation to make things practical.

$$\mathcal{L}(\delta \mathbf{w}, \lambda_1, \lambda_2) = \frac{1}{2} \delta \mathbf{w}^\top \mathbf{H} \, \delta \mathbf{w} + \lambda_1 \left( \mathbf{e}_{q_1}^\top \delta \mathbf{w} + w_{q_1} \right) + \lambda_2 \left( \mathbf{e}_{q_2}^\top \delta \mathbf{w} + w_{q_2} \right). \tag{19}$$

The lagrange dual function[S1] $g'(\lambda_1, \lambda_2)$, which is the infimum of the Lagrangian in Eq. (19) with respect to $\mathbf{w}$, can be then obtained by first differentiating Eq. 19 and setting it to 0, and then substituting the obtained value of $\delta \mathbf{w}$. These steps are indicated respectively in Eq. (20) and Eq. (21) below.

$$\mathbf{H} \delta \mathbf{w} + \lambda_1 \mathbf{e}_{q_1} + \lambda_2 \mathbf{e}_{q_2} = 0 \implies \delta \mathbf{w} = -\lambda_1 \mathbf{H}^{-1} e_{q_1} - \lambda_2 \mathbf{H}^{-1} e_{q_2}. \tag{20}$$

$$g'(\lambda_1, \lambda_2) = -\frac{\lambda_1^2}{2} \mathbf{e}_{q_1}^\top \mathbf{H}^{-1} \mathbf{e}_{q_1} + \lambda_1 w_{q_1} - \frac{\lambda_2^2}{2} \mathbf{e}_{q_2}^\top \mathbf{H}^{-1} \mathbf{e}_{q_2} + \lambda_2 w_{q_2} - \lambda_1 \lambda_2 \mathbf{e}_{q_1}^\top \mathbf{H}^{-1} \mathbf{e}_{q_2}. \tag{21}$$

Comparing this with the case when a single parameter is removed, c.f. Eq. (13), we can rewrite lagrange dual function as follows,

$$g'(\lambda_1, \lambda_2) = g(\lambda_1) + g(\lambda_2) - \lambda_1 \lambda_2 \, \mathbf{e}_{q_1}^\top \mathbf{H}^{-1} \mathbf{e}_{q_2}. \tag{22}$$

We note that dual function is not exactly separable in terms of the dual variables, $\lambda_1$ and $\lambda_2$, unless the off-diagonal term in the hessian inverse corresponding to $q_1, q_2$ is zero, i.e., $[\mathbf{H}^{-1}]_{q_1 q_2} = 0$.

To maximize the dual function in Eq. (22) above, we need to solve a linear system with the lagrange multipliers $\lambda_1, \lambda_2$ as variables. The equations for this system program correspond to setting the respect partial derivatives to zero, as described in Eq. (23) below,

$$\left. \begin{aligned} \frac{\partial g'}{\partial \lambda_1} &= -\lambda_1 \mathbf{e}_{q_1}^\top \mathbf{H}^{-1} \mathbf{e}_{q_1} - \lambda_2 \mathbf{e}_{q_1}^\top \mathbf{H}^{-1} \mathbf{e}_{q_2} + w_{q_1} = 0 \\ \frac{\partial g'}{\partial \lambda_2} &= -\lambda_1 \mathbf{e}_{q_1}^\top \mathbf{H}^{-1} \mathbf{e}_{q_2} - \lambda_2 \mathbf{e}_{q_2}^\top \mathbf{H}^{-1} \mathbf{e}_{q_2} + w_{q_2} = 0 \end{aligned} \right\} \quad \text{Solve to obtain } \lambda_1^*, \lambda_2^* \tag{23}$$

Hence, it is evident that exactly solving this resulting linear system will get intractable when we consider the removal of many parameters at once.

**Pruning Direction.** As a practical approximation to this, we build the net weight update corresponding to the removal of multiple parameters by adding the optimal weight update, Eqn. (15), computed separately for each parameter that we decide to prune. However, note that we will have to apply a mask on this weight update so as to adjust for adding the weight updates considered separately. Otherwise, the weight for the pruned parameters after applying the update might not be zero. We will call this resulting weight update as the pruning direction.

---

[S1]We denote the lagrange dual function here by $g'$ instead of $g$ to avoid confusion with the notation for lagrange dual function in case of a single multiplier.

### S1.3 Cases for specific kinds of Hessian

**Optimal Brain Damage [8].** If the hessian is assumed to be diagonal, then we can write the above Eq. (16) as follows:

$$\delta L^*_{\text{OBD}} = \frac{1}{2} w_q^2 [\mathbf{H}]_{qq}. \tag{24}$$

**Magnitude Pruning.** On top of the above case, if we assume the hessian is identity[S2], then we can write the above Eq. (16) as follows:

$$\delta L^*_{\text{Mag}} = \frac{1}{2} w_q^2. \tag{25}$$

## S2 More on the related work for IHVPs

### S2.1 K-FAC

In the recent years, an approximation called K-FAC [15, 37] has been made for the Fisher that results in a more efficient application when used as a pre-conditioner or for IHVPs. Consider we have a fully-connected network with $l$ layers. If we denote the pre-activations of a layer $i$ by $\mathbf{s}_i$, we can write them as $\mathbf{s}_i = W_i \mathbf{a}_{i-1}$, where $W_i$ is the weight matrix at the $i^{\text{th}}$ layer and $a_{i-1}$ denotes the activations from the previous layer (which the $i^{\text{th}}$ layer receives as input).

By chain rule, the gradient of the objective $L$ with respect to the weights in layer $i$, is the following: $\nabla_{W_i} L = \text{vec}(\mathbf{g}_i \mathbf{a}_{i-1}^{\top})$. Here, $\mathbf{g}_i$ is the gradient of the objective with respect to the pre-activations $s_i$ of this layer, so $\mathbf{g}_i = \nabla_{s_i} L$. Using the fact that $\text{vec}(\mathbf{u}\mathbf{v}^{\top}) = \mathbf{v} \otimes \mathbf{u}$, where $\otimes$ denotes the Kronecker product, we can simplify our expression of the gradient with respect to $W_i$ as $\nabla_{W_i} L = \mathbf{a}_{i-1}^{\top} \otimes \mathbf{g}_i$.

Then, we can then write the Fisher block corresponding to layer $i$ and $j$ as follows,

$$F_{i,j} = \text{E}\left[\nabla_{W_i} L \, \nabla_{W_j} L^{\top}\right] = \text{E}\left[\left(\mathbf{a}_{i-1} \otimes \mathbf{g}_i\right)\left(\mathbf{a}_{j-1} \otimes \mathbf{g}_j\right)^{\top}\right] \overset{(a)}{=} \text{E}\left[\left(\mathbf{a}_{i-1} \otimes \mathbf{g}_i\right)\left(\mathbf{a}_{j-1}^{\top} \otimes \mathbf{g}_j^{\top}\right)\right]$$
$$\overset{(b)}{=} \text{E}\left[\mathbf{a}_{i-1}\mathbf{a}_{j-1}^{\top} \otimes \mathbf{g}_i\mathbf{g}_j^{\top}\right], \tag{26}$$

where, in (a) and (b) we have used the transpose and mixed-product properties of Kronecker product. The expectation is taken over the model's distribution as in the formulation of Fisher.

The Kronecker Factorization (K-FAC) based approximation $\widetilde{F}$ thus used by the authors can be written as,

$$\widetilde{F}_{i,j} = \text{E}\left[\mathbf{a}_{i-1}\mathbf{a}_{j-1}^{\top}\right] \otimes \text{E}\left[\mathbf{g}_i\mathbf{g}_j^{\top}\right] = \widetilde{A}_{i-1,j-1} \otimes \widetilde{G}_{i,j} \tag{27}$$

Essentially, we have moved the expectation inside and do it prior to performing the Kronecker product. As mentioned by the authors, this is a major approximation since in general the expectation of a Kronecker product is not equal to the Kronecker product of the expectations. We will refer to $\widetilde{F}$ as the Fisher matrix underlying K-FAC or the K-FAC approximated Fisher.

The advantage of such an approximation is that it allows to compute the inverse of K-FAC approximated Fisher quite efficiently. This is because the inverse of a Kronecker product is equal to the Kronecker product of the inverses. This implies that instead of inverting one matrix of bigger size $n_{i-1}n_i \times n_{j-1}n_j$, we need to invert two smaller matrices $\widetilde{A}_{i,j}$ and $\widetilde{G}_{i,j}$ of sizes $n_{i-1} \times n_{j-1}$ and $n_i \times n_j$ respectively (here, we have denoted the number of neurons in layer $\ell$ by $n_\ell$).

As a result, K-FAC has found several applications in the last few years in: optimization [38, 39], pruning [21, 22], reinforcement-learning [51], etc. However, an aspect that has been ignored is the accuracy of this approximation, which we discuss in Section 5.1 in the context of pruning. Besides, there are a couple more challenges associated with the Kronecker-factorization based approaches.

---

[S2]Or a constant multiple of identity, it remains equivalent to magnitude pruning

**Extending to different network types.** Another issue with K-FAC is that it only naturally exists for fully-connected networks. When one proceeds to the case of convolutional or recurrent neural networks, the Kronecker structure needs to be specially designed by making further approximations [40, 41]. Whereas, a WoodFisher based method would not suffer from such a problem.

**Application to larger networks.** Furthermore, when applied to the case of large neural networks like RESNET-50, often further approximations like the chunking of block size as we consider or channel-grouping as called by [42], or assuming spatially uncorrelated activations are anyways required.

Thus, in lieu of these aspects, we argue that WoodFisher, i.e., (empirical) Fisher used along with Woodbury-based inverse is a better alternative (also see the quantitative comparison with K-FAC in Section 5.1).

## S2.2 Other methods

**Double back-propagation.** This forms the naive way of computing the entire Hessian matrix by explicitly computing each of its entries. However, such an approach is extremely slow and would require $\mathcal{O}(d^2)$ back-propagation steps, each of which has a runtime of $\mathcal{O}(md)$, where $m$ is the size of the mini-batch considered. Thus, this cubic time approach is out of the question.

**Diagonal Hessian.** If we assume the Hessian to be diagonal, this allows us to compute the inverse very easily by simply inverting the elements of the diagonal. But, even if we use the Pearlmutter's trick [52], which lets us compute the exact Hessian-vector product in linear time, we need a total of $\mathcal{O}(d)$ such matrix-vector products to estimate the diagonal, which results in an overall quadratic time.

**Diagonal Fisher.** Diagonal estimate for the empirical Fisher is really efficient to build, since it just requires computing the average of the squared gradient across the training samples, for each dimension. If the mini-batch size is $m$, we just need $\mathcal{O}(md)$ time to build this estimate. This approach has been widely used in optimization by adaptive methods [13, 14], as well for model compression by the work called Fisher-pruning [20]. However as we show ahead, by simply paying a small additional factor of $c$ in the runtime, we can estimate the inverse and IHVPs more accurately. This leads to a performance which is significantly better than that obtained via diagonal Fisher.

**Hessian-Free methods.** Another line of work is to completely forgo the explicit computation of Hessians [34], by posing the problem of computing IHVP with a vector $\mathbf{v}$ as solving the linear system $\mathbf{Hx} = \mathbf{v}$ for $\mathbf{x}$. Such methods rely on conjugate-gradients based linear-system solvers that only require matrix-vector products, which for neural networks can be obtained via Pearlmutter's trick [52]. However, a big disadvantage of these methods is that they can require a lot of iterations to converge since the underlying Hessian matrix is typically very ill-conditioned. Further, this whole procedure would have to be repeated at least $\mathcal{O}(d)$ times to build just the diagonal of the inverse, which is the minimum needed for application in model compression.

**Neumann series expansion.** These kind of methods [35, 36] essentially exploit the following result in Eqn. (28) for matrices $A$ which have an eigen-spectrum bounded between 0 and 1, i.e., $0 < \lambda(A) < 1$.

$$A^{-1} = \sum_{i=0}^{\infty} (I - A)^i \tag{28}$$

This can be then utilized to build a recurrence of the following form,

$$A_n^{-1} \triangleq I + (I - A)A_{n-1}^{-1}, \tag{29}$$

which allows us to efficiently estimate (unbiased) IHVP's via sampling. However, an important issue here is the requirement of the eigen-spectrum to be between 0 and 1, which is not true by default for the Hessian. This implies that we further need to estimate the largest absolute eigenvalue (to scale) and the smallest negative eigenvalue (to shift). Hence, requiring the use of the Power method which adds to the cost. Further, the Power method might not be able to return the smallest negative eigenvalue at all, since when applied to the Hessian (or its inverse) it would yield the eigenvalue with the largest magnitude (or smallest magnitude).

**Woodbury-based methods.** In prior work, Woodbury-based inverse has been considered for the case of a one-hidden layer neural network in Optimal Brain Surgeon (OBS, [10, 25]), where the analytical expression of the Hessian can be written as an outer product of gradients. An extension of this approach to deeper networks, called L-OBS, was proposed in [33], by defining separate layer-wise objectives, and was applied to carefully-crafted blocks at the level of neurons. Our approach via empirical Fisher is more general, and we show ahead experimentally that it yields better approximations at scale (Figure S1).

To facilitate a consistent comparison with L-OBS, we consider one-shot pruning of RESNET-50 on IMAGENET, and evaluate the performance in terms of top-5 accuracy as reported by the authors. (Besides this figure, all other results for test-accuracies are top-1 accuracies.) Here, all the layers are pruned to equal amounts, and so we first compare it with WoodFisher independent (layerwise). Further, in comparison to L-OBS, our approach also allows to automatically adjust the sparsity distributions. Thus, we also report the performance of WoodFisher joint (global) The resulting plot is illustrated in Figure S1, where we find that both independent and joint WoodFisher outperform L-OBS at all sparsity levels, and yield improvements of up to $\sim 3.5\%$ and $20\%$ respectively in test accuracy over L-OBS at the same sparsity level of $65\%$.

Figure S1: Top-5 test accuracy comparison of L-OBS and WoodFisher on IMAGENET for RESNET-50.

# S3   Visual tour detailed

(a) First-layer sub matrices averaged across over diagonal blocks of $6144 \times 6144$ for illustration purposes.

(b) Second-layer sub-matrices.

(c) Third-layer sub matrices.

Figure S2: Hessian and empirical Fisher blocks for CIFARNET ($3072 \to 16 \to 64 \to 10$) on the diagonal corresponding to different layers when trained on CIFAR10. Figures have been smoothened slightly with a Gaussian kernel for better visibility. Both Hessian and empirical Fisher have been estimated over a batch of $100$ examples in all the figures. Hessian blocks are in the left-column, while empirical Fisher blocks are displayed in the right-column.

## S3.1   All figures for Hessian and empirical Fisher comparison on CIFARNET

We consider a fully connected network, CIFARNET, with two hidden layers. Since, CIFAR10 consists of $32 \times 32$ RGB images, so we adapt the size of network as follows: $3072 \to 16 \to 64 \to 10$. Such a size is chosen for computational reasons, as the full Hessian exactly is very expensive to compute.

We follow a commonly used SGD-based optimization schedule for training this network on CIFAR10, with a learning rate $0.05$ which is decayed by a factor of 2 after every 30 epochs, momentum $0.9$, and train it for a total of 300 epochs. The checkpoint with best test accuracy is used as a final model, and

(a) Cross-matrices between first-layer and third-layer averaged across over blocks of $6144 \times 640$ for illustration purposes.

(b) Cross-matrices between second-layer and third-layer.

Figure S3: **Off-Diagonal** Hessian and empirical Fisher blocks for CIFARNET ($3072 \rightarrow 16 \rightarrow 64 \rightarrow 10$) corresponding to different layers when trained on CIFAR10. Figures have been smoothened slightly with a Gaussian kernel for better visibility. Both Hessian and empirical Fisher have been estimated over a batch of 100 examples in all the figures. Hessian blocks are in the left-column, while empirical Fisher blocks are displayed in the right-column.

this test accuracy[S3] is $41.8\%$. However, this low test accuracy is not a concern for us, as we are more interested in investigating the structures of the Hessian and the empirical Fisher matrices.

The plots in the Figure S2 illustrate the obtained matrices for the diagonal sub-matrices corresponding to the first, second, and the third layers. We observe that empirical Fisher possesses essentially the same structure as observed in the Hessian. Further, Figure S3 presents the result for the off-diagonal or cross blocks of these two matrices, where also we find a similar trend.

Thus, we conclude that the empirical Fisher shares the structure present in the Hessian matrix.

## S3.2  Across different stages of training

While previously we compared the Hessian and the empirical Fisher at convergence, in this section our aim is to show that such a trend can be observed across various stages of training, including as early as $0.5$ epochs.

For our experimental setup, we first consider a fully-connected network trained on the MNIST digit recognition dataset. This fully-connected network has two hidden layers of size $40$ and $20$. MNIST consists of $28 \times 28$ grayscale images for the digits $0 - 9$. Thus the overall shape of this network can be summarized as $784 \rightarrow 40 \rightarrow 20 \rightarrow 10$.

Note, here our purpose is not to get the best test accuracy, but rather we would like to inspect the structures of the Hessian and the empirical Fisher matrices. As a result, we choose the network with a relatively small size so as to exactly compute the full Hessian via double back-propagation. We use stochastic gradient descent (SGD) with a constant learning rate of $0.001$ and a momentum of $0.5$ to

---

[S3]In fact, this CIFARNET model with $41.8\%$ test accuracy, is far from ensuring that the model and data distribution match, yet the empirical Fisher is able to faithfully capture the structure of the Hessian.

(a) **At 0.5 epochs.** Test accuracy at this stage is 63.9%.

(b) **At 5 epochs.** Test accuracy at this stage is 85.5%.

(c) **At 50 epochs.** Test accuracy at this stage is 90.6%.

Figure S4: Last-layer Hessian and empirical Fisher blocks for MLPNET $(784 \rightarrow 40 \rightarrow 20 \rightarrow 10)$ at different points of training on MNIST. Both Hessian and empirical Fisher have been estimated over a batch of 64 examples in all the figures. Hessian blocks are in the left-column, while empirical Fisher blocks are displayed in the right-column.

train this network. The training set was subsampled to contain 5000 examples in order to prototype faster, and the batch size used during optimization was 64.

In Figure S4, we compare the last-layer sub-matrices of sizes $200 \times 200$ for both Hessian and empirical Fisher at different stages of training. We see that both these matrices share a significant amount of similarities in their structure. In other words, if we were to compute say the correlation or cosine similarity between the two matrices, it would be quite high. In these plots, the number of samples used to build the estimates of Hessian and empirical Fisher was 16, and similar trends can be observed if more samples are taken.

Thus, we can establish that the empirical Fisher shares the same underlying structure as the Hessian, even at early stages of the training, where theoretically the model and data distribution still do not match.

## S4 Experimental Details

### S4.1 Pruning schedule

We use Stochastic Gradient Descent (SGD) as an optimizer during gradual pruning, with a learning rate = 0.005, momentum = 0.9, and weight decay = 0.0001. We run the overall procedure for 100 epochs, similar to other works [27].Retraining happens during this procedure, i.e., in between the pruning steps and afterwards, as commonly done when starting from a pre-trained model.

For RESNET-50, we carry out pruning steps from epoch 1, at an interval of 5 epochs, until epoch $40^{S4}$. Once the pruning steps are over, we decay the learning rate in an exponential schedule from epoch 40 to 90 by a factor of 0.6 every 6 epochs.

For MOBILENETV1, we carry out pruning steps from epoch 4, at an interval of 4 epochs, until epoch 24. Once the pruning steps are over, we decay the learning rate in an exponential schedule from epoch 30 to 100 by a factor of 0.92 every epoch.

The amount by which to prune at every step is calculated based on the polynomial schedule, suggested in [19], with the initial sparsity percentage set to 0.05.

The same pruning schedule is followed for all: WoodFisher, Global Magnitude, and Magnitude. In fact, this whole gradual pruning procedure was originally made for magnitude pruning, and we simply replaced the pruner by WoodFisher.

### S4.2 WoodFisher Hyperparameters

The hyperparameters for WoodFisher are summarized in the Table S1. Fisher subsample size refers to the number of outer products considered for empirical Fisher. Fisher mini-batch size means the number of samples over which the gradients are averaged, before taking an outer product for Fisher. This was motivated from computation reasons, as this allows us to see a much larger number of data samples, at a significantly reduced cost.

The chunk size refers to size of diagonal blocks based on which the Hessian (and its inverse) are approximated. For RESNET-50, we typically use a chunk size of 2000, while for MOBILENETV1 we use larger chunk size of 10,000 since the total number of parameters is less in the latter case (allowing us to utilize a larger chunk size).

A thorough ablation study is carried out with respect to these hyperparameters for WoodFisher in Section S5.1.

| Model | Sparsity (%) | Fisher | | Chunk size | Batch Size |
|---|---|---|---|---|---|
| | | subsample size | mini-batch size | | |
| RESNET-50 | 80.00 | 400 | 400 | 2000 | 256 |
| | 90.00 | 400 | 400 | 2000 | 180 |
| | 95.00 | 400 | 400 | 2000 | 180 |
| | 98.00 | 400 | 400 | 1000 | 180 |
| MOBILENETV1 | 75.28 | 400 | 2400 | 10,000 | 256 |
| | 89.00 | 400 | 2400 | 10,000 | 180 |

Table S1: Detailed hyperparameters for the gradual pruning results presented in Tables S4, S6.

Note, we used a batch size of 256 or 180, depending upon whether the GPUs we were running on had 16GB memory or less. Anyhow, the same batch size was used at all the respective sparsity levels for Global Magnitude to ensure consistent comparisons.

Besides, the dampening $\lambda$ used in WoodFisher, to make the empirical Fisher positive definite, is set to $1e-5$ in all the experiments.

---

$^{S4}$All the epoch numbers are based on zero indexing.

### S4.3 Run time costs for WoodFisher pruning steps

WoodFisher indeed incurs more time during each pruning step. However, the additional time taken for these pruning steps (which are also limited in number, $\sim 6$ to $8$ in our experiments) pales in comparison to the overall 100 epochs on IMAGENET in the gradual pruning procedure.

The exact time taken in each pruning step, depends upon the value of the fisher parameters and chunk size. So more concretely, for RESNET50 this time can vary as follows e.g., $\sim 15$ minutes for fisher subsample size $= 80$, fisher subsample size $= 400$, chunk size $= 1000$ to $\sim 47$ minutes for fisher subsample size $= 160$, fisher subsample size $= 800$, chunk size $= 2000$. However, as noted in Tables S2, S3, the gains from increasing these hyperparameter values is relatively small (after a threshold), so one can simply trade-off the accuracy in lieu of time, or vice versa.

Most importantly, one has to keep in mind that compressing a particular neural network will only be done once. The **pruning cost will be amortized** over its numerous runs in the future. As a result, the extra test accuracy provided via WoodFisher is worthwhile the slightly increased running cost.

Lastly, note that, currently in our implementation we compute the inverse of the Hessian sequentially across all the blocks (or chunks). But, this *computation is amenable to parallelization* and a further speed-up can easily obtained for use in future work.

# S5    Detailed Results

## S5.1    One-shot Pruning

In all the one-shot experiments, we use the TORCHVISION models for RESNET50 and MO-BILENETV1 as dense baselines.

**RESNET50 on IMAGENET.**    Here, all layers except the first convolutional layer are pruned to the respective sparsity values in a single shot, i.e., without any re-training. Figure S5 shows how WoodFisher outperforms Global Magnitude and Magnitude with as few as 8,000 data samples. Increasing the fisher subsample size from 80 to 240 further helps a bit, and Table S2 properly investigates the effect of the fisher parameters, namely fisher subsample size and fisher mini-batch size, on the performance.

Figure S5: One-shot sparsity results for RESNET-50 on IMAGENET. In addition, we show here the effect of fisher subsample size as well as how the performance is greatly improved if we allow for recomputation of the Hessian (still no retraining). This is because the local quadratic model is only valid in a small neighbourhood (trust-region), beyond which it is not guaranteed to be accurate. The numbers corresponding to tuple of values called fisher samples refers to (fisher subsample size, fisher mini-batch size). A chunk size of 1000 was used for this experiment.

Importantly, in Figure S5, we observe that if we allow *recomputing the Hessian inverse estimate* during pruning (but without retraining), it leads to significant improvements, since the local quadratic model is valid otherwise only in a small neighbourhood or trust-region.

**MOBILENETV1 on IMAGENET.**    Similarly, we perform one-shot pruning experiments for MO-BILENETV1 on IMAGENET. Here, also we find that WoodFisher outperforms both Global Magnitude and Magnitude.

Next, Figure S6 shows the effect of fisher mini batch size, across various values of fisher subsample size, for this scenario. We notice that fisher mini-batch serves as a nice trick, which helps us take advantage of larger number of samples in the dataset at a much less cost. Further, in Table S3 presents the exact numbers for these experiments

**Effect of chunk size.**    For networks which are much larger than RESNET-20, we also need to split the layerwise blocks into smaller chunks along the diagonal. So, here we study the effect of this chunk-size on the performance of WoodFisher. We take the setting of RESNET-20 on CIFAR10 and evaluate the performance for chunk-sizes in the set, {20, 100, 1000, 5000, 12288, 37000}. Note that, 37000 corresponds to the size of the block for the layer with the most number of parameters. Thus, this would correspond to taking the complete blocks across all the layers.

| Sparsity (%) | Fisher mini-batch size | Dense: Top-1 accuracy (%) | Pruned: Top-1 accuracy (%) | |
|---|---|---|---|---|
| | | | Fisher subsample size | |
| | | | 80 | 160 |
| 30 | 1 | | $55.81 \pm 3.28$ | $57.53 \pm 1.62$ |
| | 30 | | $75.17 \pm 0.09$ | $75.26 \pm 0.17$ |
| | 100 | | $75.73 \pm 0.03$ | $75.77 \pm 0.08$ |
| | 400 | 76.13 | $75.80 \pm 0.01$ | $75.80 \pm 0.04$ |
| | 800 | | $75.74 \pm 0.04$ | $75.71 \pm 0.08$ |
| | 2400 | | $75.76 \pm 0.04$ | $75.72 \pm 0.06$ |
| 50 | 1 | | $48.76 \pm 4.95$ | $51.23 \pm 3.11$ |
| | 30 | | $73.02 \pm 0.09$ | $73.27 \pm 0.25$ |
| | 100 | | $73.70 \pm 0.13$ | $73.80 \pm 0.08$ |
| | 400 | 76.13 | $73.66 \pm 0.02$ | $73.73 \pm 0.02$ |
| | 800 | | $73.43 \pm 0.08$ | $73.47 \pm 0.06$ |
| | 2400 | | $73.32 \pm 0.10$ | $73.30 \pm 0.04$ |

Table S2: Effect of fisher subsample size and fisher mini-batch size on one-shot pruning performance of WoodFisher, for RESNET-50 on IMAGENET. A chunk size of 1000 was used for this experiment. The resutlts are averaged over three seeds.

Figure S6: Effect of fisher mini batch size, across various values of fisher subsample size, on the one-shot pruning performance of WoodFisher for MOBILENETV1 on IMAGENET.

Figures S7 and S8 illustrate the impact of the block sizes used on the performance of WoodFisher in joint and independent mode respectively. We observe that performance of WoodFisher increases monotonically as the size of the blocks (or chunk-size) is increased, for both the cases. This fits well with our expectation that a large chunk-size would lead to a more accurate estimation of the inverse. However, it also tells us that even starting from blocks of size as small as 100, there is a significant gain in comparison to magnitude pruning.

**Effect of the dampening parameter.** Regarding $\lambda$, we selected a small value so that the Hessian is not dominated by the dampening. We note that the algorithm is largely insensitive to this dampening value, Fig S9.

| Sparsity (%) | Fisher mini-batch size | Dense: Top-1 accuracy (%) | Pruned: Top-1 accuracy (%) | |
|---|---|---|---|---|
| | | | Fisher subsample size | |
| | | | 160 | 400 |
| 10 | 1 | | $43.11 \pm 4.34$ | $41.99 \pm 6.75$ |
| | 10 | | $66.86 \pm 0.79$ | $67.69 \pm 0.53$ |
| | 30 | 71.76 | $70.88 \pm 0.12$ | $70.89 \pm 0.13$ |
| | 100 | | $71.56 \pm 0.05$ | $71.59 \pm 0.14$ |
| | 400 | | $71.75 \pm 0.04$ | $71.79 \pm 0.05$ |
| | 2400 | | $71.79 \pm 0.01$ | $71.77 \pm 0.08$ |
| 30 | 1 | | $38.60 \pm 3.76$ | $38.60 \pm 6.05$ |
| | 10 | | $62.40 \pm 1.03$ | $63.54 \pm 0.74$ |
| | 30 | 71.76 | $66.90 \pm 0.11$ | $66.92 \pm 0.17$ |
| | 100 | | $67.55 \pm 0.12$ | $67.71 \pm 0.08$ |
| | 400 | | $67.88 \pm 0.06$ | $67.96 \pm 0.12$ |
| | 2400 | | $67.88 \pm 0.05$ | $67.99 \pm 0.06$ |
| 50 | 1 | | $17.64 \pm 1.62$ | $18.25 \pm 2.96$ |
| | 10 | | $28.91 \pm 1.30$ | $31.75 \pm 0.74$ |
| | 30 | 71.76 | $33.71 \pm 0.42$ | $32.10 \pm 0.42$ |
| | 100 | | $32.15 \pm 1.70$ | $31.37 \pm 0.34$ |
| | 400 | | $32.30 \pm 0.40$ | $31.46 \pm 0.13$ |
| | 2400 | | $32.39 \pm 0.67$ | $32.06 \pm 0.65$ |

Table S3: Effect of fisher subsample size and fisher mini-batch size on one-shot pruning performance of WoodFisher, for MOBILENETV1 on IMAGENET. A chunk size of $10,000$ was used for this experiment. The results are averaged over three seeds.

Figure S7: Effect of chunk size on one-shot sparsity results of WoodFisher **joint** for RESNET-20 on CIFAR10.

Figure S8: Effect of chunk size on one-shot sparsity results of WoodFisher **independent** for RESNET-20 on CIFAR10.

Figure S9: Effect of the dampening $\lambda$ on one-shot pruning results of WoodFisher (RESNET-20, CIFAR10) (avg over 4 seeds). As one would expect, the lower dampening value of $1e-5$ performs slightly better on average than the other values. This also highlights that the performance of WoodFisher is insensitive to the dampening $\lambda$.

## S5.2 Gradual Pruning

All the sparsity percentages are with respect to the weights of all the layers present, as none of the methods prune batch-norm parameters, for consistent comparisons.

**RESNET-50.** First of all, in Table S4 we present the full results for gradual pruning RESNET-50, where we also include results at $98\%$ sparsity as well as multiple results for STR [27] around the target sparsity levels.

| Method | Top-1 accuracy (%) | | Relative Drop | Sparsity | Remaining |
|---|---|---|---|---|---|
| | Dense ($D$) | Pruned ($P$) | $100 \times \frac{(P-D)}{D}$ | (%) | # of params |
| DSR [49] | 74.90 | 71.60 | -4.41 | 80.00 | 5.10 M |
| Incremental [19] | 75.95 | 74.25 | -2.24 | 73.50 | 6.79 M |
| DPF [24] | 75.95 | 75.13 | -1.08 | 79.90 | 5.15 M |
| GMP [18] | 76.69 | 75.33 | -1.77 | 80.00 | 5.12 M |
| GMP + LS [18] | 76.69 | 75.58 | -1.44 | 79.90 | 5.15 M |
| Variational Dropout [46] | 76.69 | 75.28 | -1.83 | 80.00 | 5.12 M |
| RIGL + ERK [45] | 76.80 | 75.10 | -2.21 | 80.00 | 5.12 M |
| SNFS + LS [23] | 77.00 | 74.90 | -2.73 | 80.00 | 5.12 M |
| STR [27] | 77.01 | 76.19 | -1.06 | 79.55 | 5.22 M |
| Global Magnitude | 77.01 | 76.60 | -0.53 | 80.00 | 5.12 M |
| DNW [50] | 77.50 | 76.20 | -1.67 | 80.00 | 5.12 M |
| **WoodFisher** | 77.01 | **76.73** | **-0.36** | 80.00 | 5.12 M |
| GMP [18] | 76.69 | 73.75 | -3.83 | 90.00 | 2.56 M |
| GMP + LS [18] | 76.69 | 73.91 | -3.62 | 90.00 | 2.56 M |
| Variational Dropout [46] | 76.69 | 73.84 | -3.72 | 90.27 | 2.49 M |
| RIGL + ERK [45] | 76.80 | 73.00 | -4.94 | 90.00 | 2.56 M |
| SNFS + LS [23] | 77.00 | 72.90 | -5.32 | 90.00 | 2.56 M |
| STR [27] | 77.01 | 74.73 | -2.96 | 87.70 | 3.14 M |
| STR [27] | 77.01 | 74.31 | -3.51 | 90.23 | 2.49 M |
| Global Magnitude | 77.01 | 75.09 | -2.49 | 90.00 | 2.56 M |
| DNW [50] | 77.50 | 74.00 | -4.52 | 90.00 | 2.56 M |
| **WoodFisher** | 77.01 | **75.26** | **-2.27** | 90.00 | 2.56 M |
| GMP [18] | 76.69 | 70.59 | -7.95 | 95.00 | 1.28 M |
| Variational Dropout [46] | 76.69 | 69.41 | -9.49 | 94.92 | 1.30 M |
| Variational Dropout [46] | 76.69 | 71.81 | -6.36 | 94.94 | 1.30 M |
| RIGL + ERK [45] | 76.80 | 70.00 | -8.85 | 95.00 | 1.28 M |
| DNW [50] | 77.01 | 68.30 | -11.31 | 95.00 | 1.28 M |
| STR [27] | 77.01 | 70.97 | -7.84 | 94.80 | 1.33 M |
| STR [27] | 77.01 | 70.40 | -8.58 | 95.03 | 1.27 M |
| Global Magnitude | 77.01 | 71.65 | -6.96 | 95.00 | 1.28 M |
| **WoodFisher** | 77.01 | **72.16** | **-6.30** | 95.00 | 1.28 M |
| GMP + LS [18] | 76.69 | 57.90 | -24.50 | 98.00 | 0.51 M |
| Variational Dropout [46] | 76.69 | 64.52 | -15.87 | 98.57 | 0.36 M |
| DNW [50] | 77.01 | 58.20 | -24.42 | 98.00 | 0.51 M |
| STR [27] | 77.01 | 61.46 | -20.19 | 98.05 | 0.50 M |
| STR [27] | 77.01 | 62.84 | -18.40 | 97.78 | 0.57 M |
| Global Magnitude | 77.01 | 64.17 | -16.67 | 98.00 | 0.51 M |
| **WoodFisher** | 77.01 | **65.47** | **-15.08** | 98.00 | 0.51 M |

Table S4: Comparison of WoodFisher gradual pruning results with the state-of-the-art approaches. LS denotes label smoothing, and ERK denotes the Erdős-Renyi Kernel.

Next, Gale et al. [44] also report the results for GMP and VD when run for an extended duration ($\sim 2\times$ longer compared to other methods), however, to be fair we do not compare them with other methods presented in Table S4. Nevertheless, even under such an extended pruning scenario, the final test accuracy of their models are less than that obtained by running WoodFisher for half the number of epochs. Further, unlike [44], WoodFisher does not require industry-scale extensive hyperparameter tuning.

**Additional comparisons with sparsity profile from [44].** We also carry out an additional comparison against the gradual magnitude pruning (GMP) baseline of [44]. Here, the authors show that by keeping the first convolutional layer dense, pruning the last fully-connected layer to $80\%$, and then pruning rest of the layers in the network to the desired amount, the performance of magnitude pruning is significantly improved and serves as a state-of-the-art. Thus, we run WoodFisher-*independent* in this exact setting and compare it with GMP when rest of the layers are pruned equally to $90\%$. We find that WoodFisher still outperforms GMP, even though here we do not use our automatically obtained sparsity distribution, and the results for this comparison can be found in Table S5

| | Top-1 accuracy (%) | | Relative Drop | Sparsity | Remaining |
|---|---|---|---|---|---|
| Method | Dense ($D$) | Pruned ($P$) | $100 \times \frac{(P-D)}{D}$ | (%) | # of params |
| GMP | 77.01 | 75.07 | -2.52 | 89.1 | 2.79 M |
| **WoodFisher** | 77.01 | **75.23** | **-2.31** | 89.1 | 2.79 M |

Table S5: Comparison of WoodFisher and magnitude pruning **with the same layer-wise sparsity targets** as used in [44] for RESNET50 on IMAGENET. Namely, this involves skipping the first convolutional layer, pruning the last fully connected layer to $80\%$ and the rest of the layers equally to $90\%$. WoodFisher also outperforms magnitude pruning in this setting, showing that using second-order information is helpful even with fixed sparsity targets.

**MOBILENETV1.** MobileNets [53] are a class of parameter-efficient networks designed for mobile applications, and so is commonly used as a test bed to ascertain generalizability of unstructured pruning methods. In particular, we consider the gradual pruning setting as before on MOBILENETV1 which has $\sim 4.2M$ parameters. Following STR [27], we measure the performance on two sparsity levels: $75\%$ and $90\%$ and utilize their pre-trained dense model for fair comparisons. Table S6 shows the results for WoodFisher and global magnitude along with the methods mentioned in STR. Note that the Incremental baseline from [19] keeps first convolutional and the important depthwise convolutional layers dense. However, in an aim to be network-agnostic, we let the global variant of WoodFisher to automatically adjust the sparsity distributions across the layers. Nevertheless, we observe that WoodFisher outperforms [19] as well as the other baselines: STR and global magnitude, in each of the sparsity regimes.

| | Top-1 accuracy (%) | | Relative Drop | Sparsity | Remaining |
|---|---|---|---|---|---|
| Method | Dense ($D$) | Pruned ($P$) | $100 \times \frac{(P-D)}{D}$ | (%) | # of params |
| Incremental [19] | 70.60 | 67.70 | -4.11 | $74.11^{\alpha}$ | 1.09 M |
| STR [27] | 72.00 | 68.35 | -5.07 | 75.28 | 1.04 M |
| Global Magnitude | 72.00 | 69.90 | -2.92 | 75.28 | 1.04 M |
| **WoodFisher** | 72.00 | **70.09** | **-2.65** | 75.28 | 1.04 M |
| Incremental [19] | 70.60 | 61.80 | -12.46 | $89.03^{\alpha}$ | 0.46 M |
| STR [27] | 72.00 | 62.10 | -13.75 | 89.01 | 0.46 M |
| Global Magnitude | 72.00 | 63.02 | -12.47 | 89.00 | 0.46 M |
| **WoodFisher** | 72.00 | **63.87** | **-11.29** | 89.00 | 0.46 M |

Table S6: Comparison of WoodFisher gradual pruning results for **MobileNetV1 on ImageNet** in $75\%$ and $90\%$ sparsity regime. ($^{\alpha}$) next to Incremental [19] is to highlight that the first convolutional and all depthwise convolutional layers are kept dense, unlike the other shown methods. The obtained sparsity distribution and other details can be found in the section S8.2.

## S5.3 A deeper look into gradual pruning

**What goes on during gradual pruning?** Next, to give some further insights into these results, we illustrate in Figures S10 and S11 how WoodFisher and global magnitude pruning behave during the course of gradual pruning. We observe that after almost every pruning step, WoodFisher outperforms global magnitude, and even with substantial retraining in between and after, global magnitude fails to match with WoodFisher, in terms of eventual performance. This shows the benefit of using the second order information via WoodFisher to perform superior pruning steps.

(a) Target sparsity: 80%

(b) Target sparsity: 90%

(c) Target sparsity: 95%

(d) Target sparsity: 98%

Figure S10: The course of gradual pruning with points annotated by the corresponding sparsity amounts, for **RESNET-50 on IMAGENET** across the different sparsity regimes.

(a) Target sparsity: 75.3%

(b) Target sparsity: 89%

Figure S11: The course of gradual pruning with points annotated by the corresponding sparsity amounts, for **MOBILENETV1 on IMAGENET** across the different sparsity regimes.

Figure S12: Comparison of the pruning phase during gradual pruning for WoodFisher, Global Magnitude, and Magnitude when compressing MOBILENETV1 to $60\%$ on IMAGENET. The labels on the line plot indicate the corresponding sparsity level. We observe that after each pruning step WoodFisher outperforms both Global Magnitude and Magnitude.

Besides, magnitude pruning, which prunes all layers equally performs even worse, and Figure S12 showcases the comparison between pruning steps for all the three: WoodFisher, Global Magnitude, and Magnitude. Such a trend is consistent and this is why we omit the results for magnitude pruning, and instead compare mostly with global magnitude.

## S5.4 Results averaged over multiple runs

| Method | Top-1 accuracy (%) | | Relative Drop | Sparsity | Remaining |
|---|---|---|---|---|---|
| | Dense ($D$) | Pruned ($P$) | $100 \times \frac{(P-D)}{D}$ | (%) | # of params |
| DSR [49] | 74.90 | 71.60 | -4.41 | 80.00 | 5.10 M |
| Incremental [19] | 75.95 | 74.25 | -2.24 | 73.50 | 6.79 M |
| DPF [24] | 75.95 | 75.13 | -1.08 | 79.90 | 5.15 M |
| GMP + LS [44] | 76.69 | 75.58 | -1.44 | 79.90 | 5.15 M |
| Variational Dropout [46] | 76.69 | 75.28 | -1.83 | 80.00 | 5.12 M |
| RIGL + ERK [45] | 76.80 | 75.10 | -2.21 | 80.00 | 5.12 M |
| SNFS + LS [23] | 77.00 | 74.90 | -2.73 | 80.00 | 5.12 M |
| STR [27] | 77.01 | 76.19 | -1.06 | 79.55 | 5.22 M |
| Global Magnitude | 77.01 | 76.59 | -0.55 | 80.00 | 5.12 M |
| DNW [50] | 77.50 | 76.20 | -1.67 | 80.00 | 5.12 M |
| **WoodFisher** | 77.01 | **76.76** | **-0.32** | 80.00 | 5.12 M |
| GMP + LS [44] | 76.69 | 73.91 | -3.62 | 90.00 | 2.56 M |
| Variational Dropout [46] | 76.69 | 73.84 | -3.72 | 90.27 | 2.49 M |
| RIGL + ERK [45] | 76.80 | 73.00 | -4.94 | 90.00 | 2.56 M |
| SNFS + LS [23] | 77.00 | 72.90 | -5.32 | 90.00 | 2.56 M |
| STR [27] | 77.01 | 74.31 | -3.51 | 90.23 | 2.49 M |
| Global Magnitude | 77.01 | 75.15 | -2.42 | 90.00 | 2.56 M |
| DNW [50] | 77.50 | 74.00 | -4.52 | 90.00 | 2.56 M |
| **WoodFisher** | 77.01 | **75.21** | **-2.34** | 90.00 | 2.56 M |
| GMP [44] | 76.69 | 70.59 | -7.95 | 95.00 | 1.28 M |
| Variational Dropout [46] | 76.69 | 69.41 | -9.49 | 94.92 | 1.30 M |
| Variational Dropout [46] | 76.69 | 71.81 | -6.36 | 94.94 | 1.30 M |
| RIGL + ERK [45] | 76.80 | 70.00 | -8.85 | 95.00 | 1.28 M |
| DNW [50] | 77.01 | 68.30 | -11.31 | 95.00 | 1.28 M |
| STR [27] | 77.01 | 70.97 | -7.84 | 94.80 | 1.33 M |
| STR [27] | 77.01 | 70.40 | -8.58 | 95.03 | 1.27 M |
| Global Magnitude | 77.01 | 71.72 | -6.87 | 95.00 | 1.28 M |
| **WoodFisher** | 77.01 | **72.12** | **-6.35** | 95.00 | 1.28 M |
| GMP + LS [44] | 76.69 | 57.90 | -24.50 | 98.00 | 0.51 M |
| Variational Dropout [46] | 76.69 | 64.52 | -15.87 | 98.57 | 0.36 M |
| DNW [50] | 77.01 | 58.20 | -24.42 | 98.00 | 0.51 M |
| STR [27] | 77.01 | 61.46 | -20.19 | 98.05 | 0.50 M |
| STR [27] | 77.01 | 62.84 | -18.40 | 97.78 | 0.57 M |
| Global Magnitude | 77.01 | 64.28 | -16.53 | 98.00 | 0.51 M |
| **WoodFisher** | 77.01 | **65.55** | **-14.88** | 98.00 | 0.51 M |

Table S7: Comparison of WoodFisher gradual pruning results with the state-of-the-art approaches. WoodFisher and Global Magnitude **results are averaged over two runs**. LS denotes label smoothing, and ERK denotes the Erdős-Renyi Kernel.

## S6 FLOPS

It is interesting to consider the actual speedup which can be obtained via these methods, as the total theoretical FLOP (floating point operation) counts can be lower for methods such as STR, for the same overall sparsity budget. Roughly, this is because STR leads to sparsity profiles that are relatively more "uniform" across layers, whereas WoodFisher and Global Magnitude may in theory arbitrarily re-distribute sparsity across layers. (In practice, we note that the sparsity profiles generated by these methods do correlate layer sparsity with the number of parameters in the layer.)

To test speed-up, we use the inference framework of [28], which supports efficient execution of unstructured sparse convolutional models on CPUs. We execute their framework on an Amazon EC2 c4.8xlarge instance with an 18-core Intel Haswell CPU. Sparse models are exported and executed through a modified version of the ONNX Runtime [54]. Experiments are averaged over 10 runs, and have low variance. The full results are given in Table S8.

We briefly summarize the results as follows. First, note that all methods obtain speedup relative to the dense baseline, with speedup being correlated with increase in sparsity. At the same time, the WoodFisher variants (WF) tend to have higher inference time, but also higher accuracy for the same

|  | Inference Time (ms) | | Top-1 Acc. |
| Compression | Batch 1 | Batch 64 | |
|---|---|---|---|
| Dense | 7.1 | 296 | 77.01% |
| STR-81.27% | 5.6 | 156 | 76.12% |
| **WF-Joint-80%** | 6.3 | 188 | 76.73% |
| STR-90.23% | 3.8 | 144 | 74.31% |
| **WF-Uniform-89.1%** | 4.3 | 157 | 75.23% |
| **WF-Joint-90%** | 5.0 | 151 | 75.26% |

Table S8: Comparison of inference times at batch sizes 1 and 64 for various sparse models, executed on the framework of [28], on an 18-core Haswell CPU. The table also contains the Top-1 Accuracy for the model on the ILSVRC validation set.

sparsity budget. A good comparison point is at $\sim$90% global sparsity, where WF-Joint has 0.95% higher Top-1 accuracy, for a 1.2ms difference in inference time at batch size 1. However, here the WF-Uniform-89.1% model[S5] offers a better trade-off, with similar accuracy difference, but only 0.5ms difference in inference time. We note that this latter model has lower overall sparsity than WF-Joint, since it does not prune the first and last layers, following the recipe from [18]. Generally, we found that WoodFisher models tend to be (significantly) more accurate, but that they have higher inference times. This is reasonable, since the method does not optimize for FLOPs in any way.

## S7   Sampled Fisher

|  |  | Top-1 accuracy (%) | |
| Sparsity | Dense | empirical WoodFisher | sampled WoodFisher |
|---|---|---|---|
| 20% |  | $76.10 \pm 0.04$ | $76.16 \pm 0.02$ |
| 40% | 76.13 | $75.22 \pm 0.07$ | $75.31 \pm 0.05$ |
| 60% |  | $69.21 \pm 0.05$ | $69.29 \pm 0.20$ |
| 70% |  | $48.35 \pm 0.22$ | $48.74 \pm 1.03$ |
| 80% |  | $5.48 \pm 0.45$ | $5.77 \pm 0.34$ |

Table S9: Comparison of one-shot pruning performance of WoodFisher, when the considered Fisher matrix is empirical Fisher or one-sample approximation to true Fisher, for RESNET-50 on IMAGENET. The results are averaged over three seeds.

The 'empirical' WoodFisher denotes the usual WoodFisher used throughout the paper. All the presented results in the paper are based on this setting. Whereas, the 'sampled WoodFisher' refers to sampling the output from the model's conditional distribution instead of using the labels, in order to compute the gradients. As a result, the latter can be utilized in an unsupervised setting.

In Table S9, we contrast the performance of these two options when performing one-shot pruning[S6] of RESNET-50 on IMAGENET in the *joint* mode of WoodFisher. We find that results for both types of approximations to Fisher (or whether we use labels or not) are in the same ballpark. The sampled WoodFisher does, however, have slightly higher variance which is expected since it is based on taking one sample from the model's distribution. Nevertheless, it implies that we can safely switch to this sampled WoodFisher when labels are not present.

Note that, ideally we would want to use the true Fisher, and none of the above approximations to it. But, computing the true Fisher would require $m$ additional back-propagation steps for each sampled $\mathbf{y}$ or $k$ more steps to compute the Jacobian[S7]. This would make the whole procedure $m\times$ or $k\times$ more expensive. Hence, the common choices are to either use $m = 1$ samples as employed in K-FAC [15] or simply switch to empirical Fisher (which we choose).

---

[S5]This refers to the WoodFisher version, where following [44], the first convolutional layer is not pruned and the last fully-connected layer is pruned to $80\%$, while the rest of layers are pruned to $90\%$.

[S6]As in the one-shot experiments, we use the RESNET-50 model from TORCHVISION as the dense baseline.

[S7]This is the case when a closed-form for $\mathbf{H}_\ell$ can be computed, like for exponential distributions.

Lastly, these results are based on the following hyperparameters: chunk size $= 1000$, fisher subsample size $= 240$, fisher mini batch-size $= 100$.

# S8 Sparsity distributions

As followed in the literature [24, 27], we prune only the weights in fully-connected and convolutional layers. This means that none of the batch-norm parameters or bias are pruned if present. The sparsity percentages in our work and others like [27] are also calculated based on this.

## S8.1 ResNet50

| Module | Fully Dense Params | Sparsity (%) | | | | | | | |
|---|---|---|---|---|---|---|---|---|---|
| | | WoodFisher | Global Magni | WoodFisher | Global Magni | WoodFisher | Global Magni | WoodFisher | Global Magni |
| | | 80% | | 90% | | 95% | | 98% | |
| Overall | 25502912 | | | | | | | | |
| Layer 1 - conv1 | 9408 | 37.04 | 37.61 | 44.97 | 45.72 | 51.65 | 51.86 | 60.63 | 60.09 |
| Layer 2 - layer1.0.conv1 | 4096 | 46.31 | 49.12 | 58.30 | 60.69 | 66.02 | 68.70 | 75.39 | 77.83 |
| Layer 3 - layer1.0.conv2 | 36864 | 68.18 | 68.69 | 79.48 | 80.19 | 86.81 | 87.54 | 93.03 | 93.45 |
| Layer 4 - layer1.0.conv3 | 16384 | 61.43 | 62.77 | 72.16 | 73.77 | 79.68 | 81.51 | 86.91 | 89.32 |
| Layer 5 - layer1.0.downsample.0 | 16384 | 56.10 | 57.78 | 66.31 | 68.08 | 74.10 | 75.67 | 81.68 | 83.82 |
| Layer 6 - layer1.1.conv1 | 16384 | 66.12 | 66.82 | 77.44 | 78.33 | 85.05 | 85.53 | 91.47 | 92.35 |
| Layer 7 - layer1.1.conv2 | 36864 | 71.19 | 71.78 | 82.65 | 83.01 | 89.15 | 89.52 | 94.19 | 94.76 |
| Layer 8 - layer1.1.conv3 | 16384 | 71.69 | 72.98 | 80.57 | 82.43 | 86.60 | 88.04 | 91.19 | 93.15 |
| Layer 9 - layer1.2.conv1 | 16384 | 60.47 | 61.04 | 73.18 | 74.16 | 81.82 | 83.04 | 89.84 | 90.91 |
| Layer 10 - layer1.2.conv2 | 36864 | 59.54 | 60.04 | 73.37 | 74.00 | 82.92 | 83.04 | 90.80 | 91.47 |
| Layer 11 - layer1.2.conv3 | 16384 | 72.05 | 73.14 | 79.29 | 80.76 | 84.31 | 86.17 | 89.09 | 91.46 |
| Layer 12 - layer2.0.conv1 | 32768 | 58.69 | 59.70 | 71.70 | 73.05 | 80.73 | 81.65 | 88.53 | 90.12 |
| Layer 13 - layer2.0.conv2 | 147456 | 71.07 | 71.44 | 83.83 | 84.42 | 90.77 | 91.20 | 95.70 | 96.25 |
| Layer 14 - layer2.0.conv3 | 65536 | 73.68 | 74.59 | 82.89 | 83.84 | 88.41 | 89.45 | 93.03 | 94.33 |
| Layer 15 - layer2.0.downsample.0 | 131072 | 80.55 | 81.24 | 88.98 | 89.66 | 93.31 | 94.04 | 96.60 | 97.22 |
| Layer 16 - layer2.1.conv1 | 65536 | 80.91 | 81.47 | 89.38 | 89.68 | 93.74 | 94.32 | 96.84 | 97.35 |
| Layer 17 - layer2.1.conv2 | 147456 | 77.50 | 77.66 | 87.38 | 87.63 | 92.82 | 92.95 | 96.57 | 96.71 |
| Layer 18 - layer2.1.conv3 | 65536 | 76.52 | 77.78 | 85.39 | 86.41 | 90.42 | 91.59 | 94.53 | 95.68 |
| Layer 19 - layer2.2.conv1 | 65536 | 72.53 | 73.08 | 84.34 | 85.05 | 90.64 | 91.40 | 95.21 | 96.06 |
| Layer 20 - layer2.2.conv2 | 147456 | 75.94 | 76.14 | 87.02 | 87.27 | 92.55 | 92.98 | 96.48 | 96.88 |
| Layer 21 - layer2.2.conv3 | 65536 | 69.96 | 70.99 | 82.19 | 83.21 | 88.65 | 89.91 | 93.77 | 95.04 |
| Layer 22 - layer2.3.conv1 | 65536 | 70.29 | 70.74 | 82.43 | 83.14 | 89.11 | 89.94 | 94.29 | 95.17 |
| Layer 23 - layer2.3.conv2 | 147456 | 72.67 | 72.75 | 84.80 | 85.04 | 91.15 | 91.53 | 95.85 | 96.23 |
| Layer 24 - layer2.3.conv3 | 65536 | 73.87 | 74.68 | 84.12 | 85.37 | 89.84 | 90.91 | 94.23 | 95.49 |
| Layer 25 - layer3.0.conv1 | 131072 | 62.86 | 63.53 | 76.43 | 77.31 | 84.79 | 85.61 | 91.73 | 92.67 |
| Layer 26 - layer3.0.conv2 | 589824 | 81.91 | 82.22 | 91.43 | 91.80 | 95.57 | 95.95 | 98.10 | 98.41 |
| Layer 27 - layer3.0.conv3 | 262144 | 72.28 | 72.95 | 84.20 | 85.00 | 90.76 | 91.48 | 95.39 | 96.13 |
| Layer 28 - layer3.0.downsample.0 | 524288 | 87.11 | 87.26 | 94.23 | 94.43 | 97.15 | 97.38 | 98.85 | 99.04 |
| Layer 29 - layer3.1.conv1 | 262144 | 85.79 | 85.99 | 93.09 | 93.43 | 96.38 | 96.76 | 98.32 | 98.64 |
| Layer 30 - layer3.1.conv2 | 589824 | 85.63 | 85.73 | 93.25 | 93.37 | 96.61 | 96.79 | 98.52 | 98.72 |
| Layer 31 - layer3.1.conv3 | 262144 | 77.65 | 78.16 | 88.41 | 89.09 | 93.59 | 94.24 | 96.99 | 97.58 |
| Layer 32 - layer3.2.conv1 | 262144 | 83.75 | 83.92 | 92.21 | 92.51 | 95.95 | 96.23 | 98.17 | 98.50 |
| Layer 33 - layer3.2.conv2 | 589824 | 84.99 | 84.97 | 93.31 | 93.42 | 96.73 | 96.94 | 98.63 | 98.86 |
| Layer 34 - layer3.2.conv3 | 262144 | 78.29 | 78.65 | 88.91 | 89.40 | 94.06 | 94.57 | 97.31 | 97.85 |
| Layer 35 - layer3.3.conv1 | 262144 | 81.17 | 81.27 | 90.86 | 91.14 | 95.15 | 95.52 | 97.86 | 98.21 |
| Layer 36 - layer3.3.conv2 | 589824 | 85.06 | 84.94 | 93.32 | 93.42 | 96.77 | 96.96 | 98.68 | 98.88 |
| Layer 37 - layer3.3.conv3 | 262144 | 80.29 | 80.71 | 89.93 | 90.42 | 94.54 | 95.08 | 97.54 | 98.01 |
| Layer 38 - layer3.4.conv1 | 262144 | 80.07 | 80.17 | 90.20 | 90.44 | 94.87 | 95.19 | 97.73 | 98.04 |
| Layer 39 - layer3.4.conv2 | 589824 | 84.99 | 84.95 | 93.24 | 93.37 | 96.75 | 96.92 | 98.65 | 98.88 |
| Layer 40 - layer3.4.conv3 | 262144 | 79.24 | 79.66 | 89.26 | 89.77 | 94.23 | 94.87 | 97.47 | 97.93 |
| Layer 41 - layer3.5.conv1 | 262144 | 75.83 | 75.99 | 87.68 | 87.91 | 93.44 | 93.81 | 97.07 | 97.48 |
| Layer 42 - layer3.5.conv2 | 589824 | 84.07 | 84.07 | 92.72 | 92.86 | 96.45 | 96.67 | 98.52 | 98.75 |
| Layer 43 - layer3.5.conv3 | 262144 | 75.90 | 76.42 | 87.00 | 87.59 | 92.85 | 93.52 | 96.74 | 97.38 |
| Layer 44 - layer4.0.conv1 | 524288 | 68.48 | 68.82 | 82.43 | 82.75 | 90.41 | 90.78 | 95.93 | 96.27 |
| Layer 45 - layer4.0.conv2 | 2359296 | 87.47 | 87.64 | 94.87 | 95.04 | 97.77 | 97.96 | 99.20 | 99.36 |
| Layer 46 - layer4.0.conv3 | 1048576 | 75.56 | 75.85 | 87.88 | 88.12 | 94.33 | 94.56 | 97.90 | 98.14 |
| Layer 47 - layer4.0.downsample.0 | 2097152 | 90.08 | 89.97 | 96.30 | 96.29 | 98.60 | 98.66 | 99.54 | 99.63 |
| Layer 48 - layer4.1.conv1 | 1048576 | 79.00 | 79.16 | 90.34 | 90.39 | 95.69 | 95.80 | 98.40 | 98.58 |
| Layer 49 - layer4.1.conv2 | 2359296 | 87.10 | 87.27 | 94.85 | 94.97 | 97.92 | 98.05 | 99.32 | 99.43 |
| Layer 50 - layer4.1.conv3 | 1048576 | 76.30 | 76.64 | 88.78 | 88.75 | 95.11 | 95.19 | 98.37 | 98.51 |
| Layer 51 - layer4.2.conv1 | 1048576 | 69.19 | 69.42 | 84.27 | 84.19 | 92.98 | 92.85 | 97.63 | 97.69 |
| Layer 52 - layer4.2.conv2 | 2359296 | 87.68 | 87.73 | 95.85 | 95.92 | 98.53 | 98.63 | 99.56 | 99.64 |
| Layer 53 - layer4.2.conv3 | 1048576 | 78.33 | 77.79 | 91.36 | 91.17 | 96.56 | 96.57 | 98.85 | 99.01 |
| Layer 54 - fc | 2048000 | 54.95 | 53.28 | 70.49 | 68.55 | 83.24 | 80.79 | 93.17 | 90.49 |

Table S10: The obtained distribution of sparsity across the layers by WoodFisher and Global Magnitude when sparsifying RESNET-50 to $80\%, 90\%, 95\%, 98\%$ levels on IMAGENET.

## S8.2   MobileNetV1

| Module | Fully Dense Params | Sparsity (%) | | | |
|---|---|---|---|---|---|
| | | WoodFisher | GlobalMagni | WoodFisher | GlobalMagni |
| Overall | 4209088 | 75.28 | | 89.00 | |
| Layer 1 | 864 | 50.93 | 51.16 | 55.56 | 57.99 |
| Layer 2 (dw) | 288 | 47.57 | 50.00 | 52.08 | 55.56 |
| Layer 3 | 2048 | 74.02 | 75.93 | 81.20 | 83.40 |
| Layer 4 (dw) | 576 | 18.75 | 21.01 | 26.04 | 30.21 |
| Layer 5 | 8192 | 60.05 | 60.79 | 73.44 | 74.34 |
| Layer 6 (dw) | 1152 | 30.30 | 31.86 | 39.84 | 43.75 |
| Layer 7 | 16384 | 58.16 | 58.69 | 73.55 | 74.29 |
| Layer 8 (dw) | 1152 | 07.64 | 07.90 | 15.02 | 17.45 |
| Layer 9 | 32768 | 65.53 | 65.94 | 80.06 | 80.71 |
| Layer 10 (dw) | 2304 | 33.64 | 35.68 | 45.70 | 48.13 |
| Layer 11 | 65536 | 67.88 | 68.36 | 82.99 | 83.45 |
| Layer 12 (dw) | 2304 | 16.02 | 15.41 | 25.43 | 27.26 |
| Layer 13 | 131072 | 76.40 | 76.71 | 88.93 | 89.28 |
| Layer 14 (dw) | 4608 | 38.26 | 38.85 | 51.22 | 51.58 |
| Layer 15 | 262144 | 80.23 | 80.33 | 92.20 | 92.40 |
| Layer 16 (dw) | 4608 | 49.87 | 51.65 | 64.11 | 65.84 |
| Layer 17 | 262144 | 79.29 | 79.58 | 91.92 | 92.04 |
| Layer 18 (dw) | 4608 | 49.80 | 51.19 | 64.37 | 66.43 |
| Layer 19 | 262144 | 77.42 | 77.66 | 90.90 | 91.14 |
| Layer 20 (dw) | 4608 | 43.40 | 44.88 | 60.31 | 61.98 |
| Layer 21 | 262144 | 74.51 | 74.67 | 89.47 | 89.65 |
| Layer 22 (dw) | 4608 | 30.71 | 31.62 | 50.11 | 51.89 |
| Layer 23 | 262144 | 71.09 | 71.15 | 87.93 | 88.18 |
| Layer 24 (dw) | 4608 | 17.12 | 18.12 | 41.71 | 44.34 |
| Layer 25 | 524288 | 80.30 | 80.42 | 92.62 | 92.70 |
| Layer 26 (dw) | 9216 | 62.96 | 64.45 | 79.37 | 82.56 |
| Layer 27 | 1048576 | 87.58 | 87.57 | 96.67 | 96.80 |
| Layer 28 (fc) | 1024000 | 61.11 | 60.69 | 79.91 | 79.27 |

Table S11: The obtained distribution of sparsity across the layers by WoodFisher and Global Magnitude when sparsifying MOBILENETV1 to 75%, 89% levels on IMAGENET.

## S9 WoodTaylor

### S9.1 Pruning at a general point

Incorporating the first-order gradient term should result in a more faithful estimate for the change in loss when pruning some parameter, as many times in practice, the gradient is not exactly zero. Particularly, in the case when pruning is being carried out repeatedly or when used in dynamic pruning methods, the gradients are likely to be further away from zero. We will refer to this resulting method as 'WoodTaylor'.

Essentially, this modifies the problem in Eq. (9) to as follows:

$$\min_{\delta \mathbf{w} \in \mathbb{R}^d} \left( \nabla_{\mathbf{w}} L^\top \delta \mathbf{w} + \frac{1}{2} \delta \mathbf{w}^\top \mathbf{H} \, \delta \mathbf{w} \right), \quad \text{s.t.} \quad \mathbf{e}_q^\top \delta \mathbf{w} + w_q = 0. \tag{30}$$

The corresponding Lagrangian can be then written as:

$$\mathcal{L}(\delta \mathbf{w}, \lambda) = \nabla_{\mathbf{w}} L^\top \delta \mathbf{w} + \frac{1}{2} \delta \mathbf{w}^\top \mathbf{H} \, \delta \mathbf{w} + \lambda \left( \mathbf{e}_q^\top \delta \mathbf{w} + w_q \right). \tag{31}$$

Taking the derivative of which with respect to $\delta \mathbf{w}$ yields,

$$\nabla_{\mathbf{w}} L + \mathbf{H} \delta \mathbf{w} + \lambda \mathbf{e}_q = 0 \implies \delta \mathbf{w} = -\lambda \mathbf{H}^{-1} \mathbf{e}_q - \mathbf{H}^{-1} \nabla_{\mathbf{w}} L. \tag{32}$$

The lagrange dual function $g(\lambda)$ can be then computed by putting the above value for $\delta \mathbf{w}$ in the Lagrangian in Eq. (31) as follows:

$$
\begin{aligned}
g(\lambda) \; = \;\; & -\lambda \nabla_{\mathbf{w}} L^\top \mathbf{H}^{-1} \mathbf{e}_q - \nabla_{\mathbf{w}} L^\top \mathbf{H}^{-1} \nabla_{\mathbf{w}} L \\
& + \frac{1}{2} \left( \lambda \mathbf{H}^{-1} \mathbf{e}_q + \mathbf{H}^{-1} \nabla_{\mathbf{w}} L \right)^\top \mathbf{H} \left( \lambda \mathbf{H}^{-1} \mathbf{e}_q + \mathbf{H}^{-1} \nabla_{\mathbf{w}} L \right) \\
& + \lambda \left( - \lambda \mathbf{e}_q^\top \mathbf{H}^{-1} \mathbf{e}_q - \mathbf{e}_q^\top \mathbf{H}^{-1} \nabla_{\mathbf{w}} L + w_q \right) \\
= \;\; & -\frac{\lambda^2}{2} \mathbf{e}_q^\top \mathbf{H}^{-1} \mathbf{e}_q - \lambda \mathbf{e}_q^\top \mathbf{H}^{-1} \nabla_{\mathbf{w}} L + \lambda w_q - \frac{1}{2} \nabla_{\mathbf{w}} L^\top \mathbf{H}^{-1} \nabla_{\mathbf{w}} L. 
\end{aligned} \tag{33}
$$

As before, maximizing with respect to $\lambda$, we obtain that the optimal value $\lambda^*$ of this lagrange multiplier as follows:

$$\lambda^* = \frac{w_q - \mathbf{e}_q^\top \mathbf{H}^{-1} \nabla_{\mathbf{w}} L}{\mathbf{e}_q^\top \mathbf{H}^{-1} \mathbf{e}_q} = \frac{w_q - \mathbf{e}_q^\top \mathbf{H}^{-1} \nabla_{\mathbf{w}} L}{[\mathbf{H}^{-1}]_{qq}}. \tag{34}$$

Note, if the gradient was 0, then we would recover the same $\lambda^*$ as in Eq. (14). Next, the corresponding optimal perturbation, $\delta \mathbf{w}^*$, so obtained is as follows:

$$\delta \mathbf{w}^* = \frac{- \left( w_q - \mathbf{e}_q^\top \mathbf{H}^{-1} \nabla_{\mathbf{w}} L \right) \mathbf{H}^{-1} \mathbf{e}_q}{[\mathbf{H}^{-1}]_{qq}} - \mathbf{H}^{-1} \nabla_{\mathbf{w}} L. \tag{35}$$

In the end, the resulting change in loss corresponding to the optimal perturbation that removes parameter $w_q$ can be written as (after some calculations[S8]),

$$\delta L^* = \frac{w_q^2}{2 \left[ \mathbf{H}^{-1} \right]_{qq}} + \frac{1}{2} \frac{\left( \mathbf{e}_q^\top \mathbf{H}^{-1} \nabla_{\mathbf{w}} L \right)^2}{\left[ \mathbf{H}^{-1} \right]_{qq}} - w_q \frac{\mathbf{e}_q^\top \mathbf{H}^{-1} \nabla_{\mathbf{w}} L}{\left[ \mathbf{H}^{-1} \right]_{qq}} - \frac{1}{2} \nabla_{\mathbf{w}} L^\top \mathbf{H}^{-1} \nabla_{\mathbf{w}} L. \tag{36}$$

Lastly, choosing the best parameter $\mathbf{w}_q$ to be removed, corresponds to one which leads to the minimum value of the above change in loss. As in Section S1.2, our pruning statistic $\rho$ in this setting can be similarly defined, in addition by excluding the last term in the above Eq. (36) since it does not involved the choice of removed parameter $q$. This is indicated in the Eq. (37) below.

$$\boxed{\rho_q = \frac{w_q^2}{2 \left[ \mathbf{H}^{-1} \right]_{qq}} + \frac{1}{2} \frac{\left( \mathbf{e}_q^\top \mathbf{H}^{-1} \nabla_{\mathbf{w}} L \right)^2}{\left[ \mathbf{H}^{-1} \right]_{qq}} - w_q \frac{\mathbf{e}_q^\top \mathbf{H}^{-1} \nabla_{\mathbf{w}} L}{\left[ \mathbf{H}^{-1} \right]_{qq}}.} \tag{37}$$

---

[S8]It's easier to put the optimal value of $\lambda^*$ in the dual function (Eq. (33)) and use duality, than substituting the optimal perturbation $\delta \mathbf{w}^*$ in the primal objective.

## S9.2    Partially trained model

First, we present the results for the case when the model is far from the optimum, and hence the gradient is not close to zero. This setting will allow us to clearly see the effect of incorporating the first-order gradient term, considered in the WoodTaylor analysis. In particular, we consider an MLPNET on MNIST, which has only been trained for 2 epochs.

Figure S13 presents the results[S9] for performing one-shot compression for various sparsity levels at this stage in the training. Similar to the results in past, we find that WoodTaylor is significantly better than magnitude or diagonal Fisher based pruning as well as the global version of magnitude pruning. But the more interesting aspect is the improvement brought about by WoodTaylor, which in fact also improves over the accuracy of the initial dense model by about $5\%$, up to sparsity levels of $80\%$.

Figure S13: Comparing one-shot sparsity results for WoodTaylor and baselines on the partially trained MLPNET on CIFAR-10.

This points towards the potential benefit of using WoodTaylor in the dynamic pruning scenario, like along with [24]. Further, we show ahead that this benefit of using WoodTaylor over WoodFisher can be observed in the pre-trained setting as well, where the gradient is close to zero (albeit smaller relative to here).

## S9.3    Pre-trained model

Next, we focus on the comparison between WoodFisher and WoodTaylor for the setting of ResNet-20 pre-trained on CIFAR10, where both the methods are used in their 'full-matrix' mode. In other words, no block-wise assumption is made, and we consider pruning only the 'layer1.0.conv1', 'layer1.0.conv2' and 'layer2.0.conv1'. In Figures S14, S15, we present the results of one-shot experiments in this setting. We observe that WoodTaylor (with damp=$1e-3$) outperforms WoodFisher (across various dampening values) for almost all levels of target sparsity. This confirms our hypothesis of factoring in the gradient term, which even in this case where the model has relatively high accuracy, can lead to a gain in performance. However, it is important to that in comparison to WoodFisher, WoodTaylor is more sensitive to the choice of hyper-parameters like the dampening value, as reflected in the Figure S14. This arises because now in the weight update, Eqn. (35), there are interactions between the Hessian inverse and gradient terms, due to which the scaling of the inverse Hessian governed by this dampening becomes more important. To give an example, in the case where damp=$1e-5$, the resulting weight update has about $10\times$ bigger norm than that of the original weight.

---

[S9]Here, the number of samples used for Fisher in both WoodTaylor and WoodFisher is 8000. A dampening of $1e-1$ was used for WoodTaylor, while WoodFisher is insensitive to dampening as discussed in the next section.

Figure S14: Comparing one-shot sparsity results for WoodTaylor and WoodFisher on CIFAR-10 for ResNet-20.

Figure S15: A simplified comparison of one-shot sparsity results for WoodTaylor and WoodFisher on CIFAR-10 for ResNet-20.

This can be easily adjusted via the dampening, but unlike WoodFisher, it is not hyper-parameter free. Also, for these experiments, the number of samples used was 50,000, which is higher in comparison to our previously used values. In order to better understand the sensitivity of WoodTaylor with respect to these hyper-parameter choices, we present an ablation study in Figure S16 that measures their effect on WoodTaylor's performance.

In the end, we conclude that incorporating the first-order term helps WoodTaylor to gain in performance over WoodFisher, however, some hyper-parameter tuning for the dampening constant might be required. Future work would aim to apply WoodTaylor in the setting of gradual pruning discussed in Section S5.3.

Figure S16: Ablation study for WoodTaylor that shows the effect of dampening and the number of samples used on the performance.