[Reviews · NeurIPS 2020]

Review 1

Summary and Contributions: This paper has two contributions. First, the author used the Sherman-Morrison equation (or in general the Woodbury equation) to efficiently approximate the Fisher information matrix F. See equation (2) and (3). Second, the author applied the approximated F to one-shot pruning and online pruning, showing better performance than state-of-the-art alternatives on CNNs. The intuition is to perform pruning on the direction where the loss landscape bends the least upwards.

Strengths: This paper has two highlights corresponding to the two contributions. First, it is very interesting to have an efficient approximation of F which can be used in broader applications. Second, the application of F in the area of neural network pruning is interesting and the authors managed to build a tool to efficiently prune the CNNs both online and offline, beating state-of-the-art records.

Weaknesses: The main weakness is that the first contribution is not new, and the authors somehow re-invented an old technique that is known in the 90s. The authors' approximation of F is based on the Sherman-Morrison formula (although they named it "WoodFisher", it did not use the more general Woodbury equality). This approximation has been known and used. See e.g. equation 16 [1]. There could be minor differences (such as the step size) but the main technique is well known. These previous contributions should be properly acknowledged and the paper has to be written in a way to apply [1] into the area of neural network pruning, with the main contribution lying on this application (the authors' second contribution). Furthermore, as the technique is not new, it may not proper to give a new name. Given that this revision will require some rewriting, I am putting a "rejection" instead of "weak rejection" here. [1] Adaptive Method of Realizing Natural Gradient Learning for Multilayer Perceptrons. Amari et al. 2000

Correctness: I could not identify any issues on the corretness.

Clarity: In terms of language, the paper is well written. In terms of the formulations, there should be more technical details in section 4, as this is the main contribution. Otherwise, the formulations are well organized.

Relation to Prior Work: See the above "weakness". I suggest the authors have more investigations in the next round of revision as the area of second-order optimization in deep learning is becoming more popular, and properly claim the contributions.

Reproducibility: Yes

Additional Feedback: Some minor comments: There are several bad-links in the paper. For example, page 3, Appendix ?? section 2. "Empirical Fisher" needs some reference mentioned. Most of the figures and tables are hard to read (font too small) There should be more technical details on how the proposed "WoodFisher" is computed after eq.(3) in terms of equations or algorithms. Page 4 "The runtime is this reduced" rewrite Table 1, add some gap between the caption and content. What do you mean by "Applicability"? Page 5, Figure S1 is not in the main text. Extend section 4 to give a detailed algorithm. --- Thank you for your rebuttal. I am willing to increase my score to "weak reject" based on that you agree to make clear the relationships with prior works, and on that this paper has as a relatively good quality in my batch. Notice it is not sufficient to merely cite [Amari et al 2000], but rather to introduce your work as an increment over these previous works. For example, at the beginning of section 3.2, it has to be written in a way to make clear what is already in these previous works, and what is new. Similarly, there are several places that need to be rewritten.


Review 2

Summary and Contributions: This paper proposes to use the Woodbury matrix identity in order to efficiently invert the matrix of the second moment of the gradients (a.k.a. "empirical Fisher"), exploiting the fact that the matrix is rank n (number of examples in dataset) with n << d (d number of parameters). This inverse is used in the context of unstructured pruning, using the OBS saliency/update, and the technique is empirically compared against other unstructured pruning methods, for one shot pruning and gradual pruning. In addition to that, usage of the 2nd moment of the gradient is motivated visually and by comparing the change predicted by the OBS model to the actual loss change. Other approximations (K-FAC) are also discussed, incorrectly in my opinion (see comments below). A large empirical ablation study for different hyperparameters is conducted, as well as a benchmark against popular competitors for one shot pruning and gradual pruning.

Strengths: -- Technical details The technique is clearly motivated, which consists in applying a well known linear algebra trick in order to improve unstructured pruning techniques. -- Empirical While I have some reservations about it (see "weaknesses"), the empirical evaluation shows a consistent improvement compared to other recent techniques, on 2 different tasks using several architectures. The ablation study (in appendix) is also interesting and would be relevant even for other techniques.

Weaknesses: --- Missing details about lambda While mentioned line 138, the dampening parameter lambda does not appear in the experimental section of the main body, and I only found a value 1e-5 in the appendix (l799). How do you select its value? I expect your final algorithm be very sensitive to lambda, since \delta_L as defined in eq.4 will select directions with smallest curvature. Another comment about lambda is that if you set it to a very large value k, then its becomes dominant compared to the eigenvalues of F, then your technique basically amounts to magnitude pruning. In that regards, it means that MP is just a special case of your technique, when using a large dampening value. Even if not mentioned, I am assuming that you perform an hyperparameter search for lambda, and I am thus not so surprised that your technique improves over MP. It is similar to adding a parameter lambda and optimize it with a hyperparameter search, e.g. using a validation set. -- Fisher minibatch size What is the motivation for using averaged gradients instead of individual gradients as in eq 1 and the next eq (not numbered)? -- Multiple hessian inverses during a pruning step It seems that you obtain an important improvement by allowing multiple recomputations of the inverse hessian. Can you clarify in which experiment you do actually use this trick? If you do use it during pruning steps in gradual pruning experiments, does it allow a fair comparison against other techniques? -- Comparison to K-FAC in fig 3 and S3.3 I suspect that you plotted B kron A instead of A kron B, that would be (at least visually) much more similar to the actual Fisher, an error that I myself do quite often when experimenting with K-FAC. Since your argument against using K-FAC is only based on this visual exploration, I think your point is thus invalid. -- Approximation quality of local quadratic model Can you provide more details about the experiment in fig 2? I read line 128 that F and H are roughly similar, up to a multiplicative constant, a statement with which I fully agree. So from the fact that the quadratic model approximates the loss quite accurately, I assume that you included this scaling constant? If so, how did you choose it ? Otherwise, what is happening here ? -- Overall My impression is that the message is a bit diluted due to much material. For instance I would completely delete the part about K-FAC, that I find not so much relevant for WoodFisher, and instead spend more space describing your conclusions about fisher minibatch size, recomputation of the Hessian during pruning, choice of rank of the Fisher, etc..

Correctness: --K-FAC I think that the K-FAC comparison is flawed. It should be corrected. -- Technical The method is correct and well motivated. -- Empirical It is always difficult to assess the empirical results of a paper, because of the tendency to always spend more time evaluating (and tuning) ones algorithm than the competitors. Do you maybe have a comment about that?

Clarity: The paper is well written, with the small drawback that there is a lot of material, thus everything is quite condensed.

Relation to Prior Work: Other related work is clearly discussed.

Reproducibility: Yes

Additional Feedback: I find a bit weird that you call your update (eq 4) optimal brain damage, since it is different from the one derived in the OBD paper. Why not call it optimal brain surgeon, as in [10] ? l88 and eq 1 are not consistent since the expectation is over the joint in eq 1 but you mention the conditional P_y|x on line 88. refs [18] and [43] are the same one. -- after author's rebuttal: I thank the author's for their additional K-FAC comparison, I am however not so confident that it is correct given the short time frame, also given the fact that there is an error in the implementation of K-FAC in the submission. In the current state, I think no (possibly false) claim should be made of superior performance of Woodbury low rank pseudo inverse vs K-FAC. Apart from that, the paper is of overall good quality. I will thus keep my score unchanged.


Review 3

Summary and Contributions: In this paper the authors introduce an approximate second-order neural network compression scheme. They use the Woodbury matrix identity to construct an approximation to the inverse of the empirical Fisher, and use it to select weights to prune from a neural network.

Strengths: -impressive results on one-shot and gradual pruning -visualizations of the various curvature matrices are very useful and informative! -analysis in fig 2 is very insightful -background was concise but a useful refresher

Weaknesses: -font sizes in fig 2 could be larger -figure reference on line 322 broken -appenfix reference on line 124 broken -compression performance comparisons using other approximate Fisher matrices (such as KFAC) for compression could be useful, but not required

Correctness: The claims appear to be correct.

Clarity: The writing is well done, balancing detail and brevity.

Relation to Prior Work: The authors compare against many relevant and competitive baselines, and do a thorough job discussing previous and related works.

Reproducibility: Yes

Additional Feedback: REBUTTAL RESPONSE: Overall I still believe this is a well done work, but I agree with the other reviewers that a more rigorous comparison to K-FAC approximations would be very useful for this paper and request the authors add these experiments to the final work in their tables/figures. ======================================================== Overall this is a thorough and well-constructed paper, which should be accepted. While it is not the first paper to consider approximate Fisher matrices for model compression (https://arxiv.org/abs/1905.05934), it uses a unique approximation (for the compression application), and explores the pros and cons of the method. The thoroughness with which it explores various individual aspects of the proposed method is impressive.


Review 4

Summary and Contributions: This paper introduces a computationally efficient way to compute second-order information of large-scale neural networks. In particular, the authors demonstrate that the inverse of the Hessian can be approximated by leveraging the relationship between the Hessian and the Fisher information matrix and applying the Woodbury matrix identity iteratively to compute the inverse of empirical Fisher matrix, which they (empirically) show provides an accurate approximation of the Hessian inverse. The authors then apply their algorithm to the problem of network pruning and achieve state-of-the-art results.

Strengths: - The paper is highly relevant to the ML and optimization communities - The authors provide a very thorough background on each topic and provide theoretical justification for the link between network pruning and the necessity of having inverse Hessian approximations (due to the Taylor expansion). I also appreciated their detail in their derivations in the appendix - The proposed algorithm builds upon prior work and is theoretically grounded - The authors provide extensive evaluations on pertinent data sets and networks for both one-shot and gradual pruning and show that their algorithm consistently achieves state-of-the-art results. The authors also show (in the Appendix) that they outperform the related work that also used WoodBury (Optimal Brain Surgeon (OBS) [10]), and this supports the validity of the author’s extensions of previous work -The authors also derive and propose a pruning variant that incorporates both first and second order terms for better pruning performance, and they provide extensive empirical evaluations showing its efficacy in practice (S9 - WoodTaylor in the appendix).

Weaknesses: Theory and hyper-parameters: I would have appreciated some high-level theoretical guidelines (not necessarily full proofs) on the quality of the approximations used at each step of the way. For example, how well can we expect the Fisher information matrix to approximate the Hessian in practice? Are there any real-world data sets where it wouldn’t be well-approximated? I think that these bounds could also shed light on the scale of the hyper-parameters used. For example, for the Block-wise Approximation, is there any way of setting the block size parameter c in advance?

Correctness: Yes, see above.

Clarity: The paper is very well-written and organized. The appendix is also very well organized and insightful, especially regarding the theoretical derivations of the pruning algorithm(s).

Relation to Prior Work: For the most part yes, however, regarding their work, the authors state that “We emphasize that a similar approach was suggested in the early work of [10] (OBS)... Our contribution is in extending this idea..” It would be great if the authors could highlight what their specific extensions are and how they improve upon prior work. As noted above in the strengths, the authors do provide empirical evidence that their work is superior to OBS in pruning (i, however, I think it would be clarifying to outline the similarities and differences and justify their claim that these extensions enable better performance relative to OBS.

Reproducibility: Yes

Additional Feedback: Update: In light of the concerns raised by the other reviewers over comparisons to K-FAC and novelty relative to prior work, I have changed my score to a 7. I still think the paper is a clear and thorough exposition that builds on prior work to combine existing techinques in a novel way to achieve SOTA pruning results -- I believe this will be appreciated by the pruning community.

[Author Response · NeurIPS 2020]

We would like to thank the reviewers for their comments, and take the opportunity to answer their questions below.

**R1: (1)** We thank the reviewer for the relevant [Amari et al., 2000] reference, which we will cite and discuss. To our knowledge, the first reference to suggest a similar procedure for approximating IHVP is (Hassibi & Stork, 1993), cited as [10], in the context of pruning. We repeatedly acknowledged (e.g. lines 49-51, 195-197) that our contribution is **not** in introducing this technique, but in considering its applicability, accuracy, and efficiency in the context of pruning modern deep networks, for which we show state-of-art results. ([10] considered pruning single-layer neural networks with at most 65 parameters on small datasets. Similarly, [Amari et al., 2000] considers single-layer networks with < 50 parameters.) Further, we examined the method's accuracy relative to recent techniques, and extended it to account for first-order information. **(2)** We are open to changing the term "WoodFisher" which we used as a mnemonic and to simplify the differentiation between variants (e.g. WoodFisher -> WoodTaylor). We are confident that changing the name and discussing [Amari et al., 2000] would not require a significant revision. We gently ask the reviewer to reconsider their score in view of this. **(3)** By applicability, we mean if the methods can be applied to any network type (see Appendix S2 for more). Also, we will provide additional technical details in Section 4 and address the other comments.

Figure 1: (a) WoodFisher vs K-FAC (MLPNet, MNIST) (b) Effect of $\lambda$: WoodFisher (ResNet-20, CIFAR10) (avg over 4 seeds)

**R2: (1)** Regarding $\lambda$, we selected a small value so that the Hessian is not dominated by the dampening. We note that the algorithm is largely insensitive to this dampening value, Fig 1b. We did not perform an exhaustive hyperparameter search for $\lambda$: we identified a small value (1e-5) which worked, and adopted it across all our experiments without further tuning. Indeed, huge $\lambda$ would approximate global magnitude pruning. But, it is not exactly the same as simply tuning this $\lambda$ alone, as that would only control the diagonal. **(2)** Fisher minibatch size: This is a practical trick which allows us to utilize more samples at the same cost. Please see Appendix S5 for ablation studies. **(3)** We only use multiple Hessian inverses for the one-shot pruning experiment on ImageNet. We did not use it in gradual pruning, so our comparisons are completely fair. **(4)** Comparison to K-FAC: that's a great catch. To address this comparison issue, we have implemented K-FAC-based pruning, and found that it is clearly outperformed by our method, even on a simple MLP example (see Figure 1a). For convolutional layers, K-FAC needs to make additional approximations, so we can expect the results to further improve. **(5)** For simplicity, we consider the scaling constant as 1 here. Fig 3 illustrates the possible cases where the quadratic model {over, under, closely} - estimates the loss based on $\lambda$. Note, pruning is independent of this scaling constant. **(6)** We believe our self-tuning is fairly limited: we simply adopted the hyperparameter combination which works best for magnitude pruning (as followed in the literature) and plugged in WoodFisher as the estimator. In fact, post-submission, we noticed that some hyperparameter values (e.g. weight decay) can be further tuned to give an additional boost to our method (at $89\%$ sparsity, MobileNetV1 results increase from $63.87 \to 64.59$, etc.). Plus, we do not use "additional tricks" such as label smoothing or cosine LR schedules, which can further help accuracy (see STR [26]). **(7)** We will carefully follow your notes on presentation.

**R3:** Thanks for the suggestions, we will correct the font sizes & the broken references. (Lines 522-524 in the Appendix contain the corrected references.) Please see the K-FAC results in Fig 1a, which we will add in the paper too.

**R4: (1)** Theoretical guarantees: When the model and data distribution match, the (true) Fisher and Hessian are indeed equivalent. Our visual analysis of the Hessian and empirical Fisher serves to showcase the approximation quality under relaxed conditions that arise in practice. Further, in the loss analysis on ResNet-20, we observe that empirical Fisher can estimate the loss in a local neighbourhood remarkably well. We are very interested in developing proper theoretical guarantees to the approximation quality in future work; the WoodTaylor variant in fact came out of this effort. **(2)** Our Hessian approximation wouldn't make sense in the interpolation setting, where the gradient at each sample is already zero. But this is an artificial example, which is unlikely to arise in practice. **(3)** Block size is calculated based on model size and CPU memory. We always seek to maximize block size; c.f. Fig. S10 for the ablation study. **(4)** Relative to OBS, we apply a similar method to modern-day networks such as ResNet50, while [10] was applied only to a single hidden layer MLP. This is made possible by the combination of techniques: block-wise chunking, mini-batch gradients in empirical Fisher to utilize more samples, and proper book-keeping to enable this implementation in PyTorch. In addition, we incorporate the first-order term ignored by [10] and most of the literature (WoodTaylor), but can be vital in the dynamic pruning setting (see results in Fig S14). Lastly, in Fig S1, we show that we outperform L-OBS [32] which is a recent attempt to make [10] practically viable by defining separate layer-wise losses.

[Meta-Review · NeurIPS 2020]

The focus of the submission is training neural networks using 2nd-order information. Particularly, the goal of the work is the approximation of the inverse of the empirical Fisher matrix as it is defined in the displayed equation under (1). The authors notice that the empirical Fisher is an average of diads (a x a^T where ^T denotes transposition) hence its inverse can be recursively computed by the Woodbury matrix identity. The resulting inverse is applied for pruning of convolutional neural networks (CNNs) and is compared against other unstructured pruning methods. Training and pruning neural networks are central problems of machine learning. While the mathematical contribution in the submission is somewhat limited (the Woodbury matrix identity is a quite standard approach in numerical analysis), the reviewers agreed that the empirical evaluation is thorough, the approach is useful and can have impact in the area of CNNs. In the final version of the paper, 1) a more rigorous comparison to K-FAC approximations is suggested to be carried out, 2) the novelty of the submission (=application in CNN pruning) should be more clearly emphasized.